# The impacts of marine-emitted halogens on OH radicals in East Asia during summer

Shidong Fan[1,2], Ying Li[1,2,3]

[1]Department of Ocean Science and Engineering, Southern University of Science and Technology, Shenzhen 518055, China
5 [2]Center for the Oceanic and Atmospheric Science at SUSTech (COAST), Southern University of Science and Technology, Shenzhen 518055, China
[3]Southern Marine Science and Engineering Guangdong Laboratory (Guangzhou) China

*Correspondence to*: Ying Li (liy66@sustech.edu.cn)

10 **Abstract.** Relationships between oceanic emissions and air chemistry are intricate and still not fully understood. For regional air chemistry, a better understanding of marine halogen emission on hydroxyl (OH) radical is crucial. The OH radical is a key species in atmospheric chemistry because it can oxidize almost all trace species in the atmosphere. In the marine atmosphere, OH levels could be significantly affected by the halogen species emitted from the ocean. However, due to the complicated interactions of halogens with OH through different pathways, it is not well understood how halogens influence 15 OH and even what the sign of the net effect is. Therefore, in this study, we aim to quantify the impact of marine-emitted halogens (including Cl, Br, and I) through different pathways on OH in the high OH season by using WRF-CMAQ model with process analysis and state-of-the-art halogen chemistry in the East Asia and near western pacific. Results show a very complicated response of the OH production rate ($P_{OH}$) to marine halogen emissions. The monthly $P_{OH}$ is generally decreased over the ocean by up to a maximum of about 10–15% in the Philippine Sea, but is increased in many nearshore areas by up 20 to about 7–9% in the Bohai Sea. In the coastal areas of southern China, the monthly $P_{OH}$ could also decrease 3–5%, but hourly values can decrease over 30% in the daytime. Analysis of the individual reactions using integrated reaction rate shows that the net change of $P_{OH}$ is controlled by the competitions of three main pathways (OH from $O_3$ photolysis, OH from $HO_2$ conversion, and OH from HOX, X=Cl, Br, I) through different halogen species. Sea spray aerosols (SSA) and inorganic iodine gases are the major species to influence the strengths of these three pathways and therefore have the most significant 25 impacts on $P_{OH}$. Both of these two types of species decrease $P_{OH}$ through physical processes, while generally increasing $P_{OH}$ through chemical processes. In the ocean atmosphere, inorganic iodine gases determine the basic pattern of $\Delta P_{OH}$ through complicated iodine chemistry which generally positively influences $P_{OH}$ near $O_3$ sources while negatively when $O_3$ experiences longer transport over the ocean. Over the continent, SSA is the controlling species and the SSA extinction effect leads to the negative $\Delta P_{OH}$ in southern China. Our results show that marine-emitted halogen species have notable impacts 30 over the ocean and potential impact on coastal atmospheric oxidation by species (SSA, inorganic iodine, and halocarbons), processes (chemistry, radiation, and deposition) and main pathways. The notable impacts of the marine-emitted halogen species on the atmospheric oxidation capacity have further implications on the lifetime of long-lived species such as $CH_4$ in

long-term and amount of air pollutants such as $O_3$ in the episodic events in East Asia and in other circumstances (e.g., different domains, regions, and emission rates).


## 1 Introduction

The hydroxyl radical is the most important daytime oxidant in the troposphere. It can oxidize almost all directly emitted gases such as CO, $CH_4$ and other volatile organic compounds (VOCs), while producing some secondary species such as $O_3$ and secondary aerosols at the same time. The primary source of OH in the troposphere is $O_3$, through the reaction of water

vapor with $O(^1D)$ which is produced from the photolysis of $O_3$. At urban areas, the photolysis of HONO is also a significant source of OH, and may be more important than the photolysis of $O_3$, especially in spring, autumn, and winter because of the very large seasonal variations of $O_3$ photolysis and humidity (e.g., Tan et al., 2019;Whalley et al., 2021;Liu et al., 2019). When there is abundant NO, as typically in the polluted continental atmosphere, peroxy radicals ($RO_2$ and $HO_2$) will be formed by the oxidation of hydrocarbons by OH, and will form OH again in the reaction with NO. This $HOx$ (=OH+$HO_2$)

cycling maintains a high OH concentration that cannot be achieved by primary sources alone. The main sinks of OH are CO and VOCs. Due to the complexity of the $HOx$ chemistry, the sources and sinks of OH are not fully understood. For example, recent studies showed that when $NOx$ concentration is very low (e.g., NO concentration less than several hundred pptv) there may be missing sources of OH (Tan et al., 2019;Rohrer et al., 2014;Lelieveld et al., 2008;Hofzumahaus et al., 2009;Lu et al., 2019a;Fuchs et al., 2013;Stone et al., 2012;Fittschen et al., 2019;Whalley et al., 2021). In addition, $HOx$ chemistry can

interact with other oxidizers in the atmosphere in specific circumstances.

In the marine atmosphere, the abundant marine-emitted halogen species have significant impacts on OH. Marine-emitted halogen could make the tropospheric $HOx$-$NOx$-$O_3$-VOCs chemistry more complex. One relevant reaction is that XO (X=Cl, Br, and I) shifts the $HOx$ balance towards OH (Saiz-Lopez and von Glasow, 2012). As a consequence, previous box-model studies usually showed positive impacts of halogen chemistry on OH (Stone et al., 2018;Whalley et al., 2010). However,

there is an opposite impact of halogens on OH as usually shown by chemical transport model (CTM) studies that halogen species will consume $O_3$ which in turn would reduce the production of OH (e.g., Sherwen et al., 2016;Stone et al., 2018). For example, Wang et al. (2021) showed that the net effect of halogen chemistry on global tropospheric $HOx$ is that both OH and $HO_2$ are reduced by 3–4%. In a box model, when the long-lived species such as $O_3$ are observation-constrained, it cannot reflect the complete influence of halogens, which probably explains the different results between box models and CTMs

(Stone et al., 2018). Therefore, special attention needs to be paid when using box models to quantify the complicated impacts of halogen species on the $HOx$-$NOx$-$O_3$-VOCs chemistry.

To present the global impact quantitatively, a more comprehensive understanding on the changes by species (sea spray aerosols (SSA), inorganic iodine, and halocarbons) and their associated processes (chemistry, radiation, and deposition) is needed in order to better explain relevant observed or modeled phenomena and their driving factors. However, the pathways

and processes by which halogens influence OH have not been well quantified in previous studies. Recent studies in understanding the impacts of halogen chemistry on OH usually focused on the two pre-described pathways (i.e., enhanced $HO_2$ conversion by XO and $O_3$ consumption by X atoms). Even though we know all the important pathways, due to their opposite impacts on OH, we need further to understand the controlling processes of these pathways in order to better explain the trend of halogen-induced $\Delta$OH in a specific circumstance. Moreover, since current estimations of marine halogen

emissions, including SSA Cl and Br ions (and their activations), inorganic iodine ($I_2$ and HOI), and very short-lived halocarbons), have large uncertainties (Carpenter et al., 2021;Ordóñez et al., 2012;Ziska et al., 2013;Inamdar et al., 2020;Lennartz et al., 2015;Zhu et al., 2019;Sekiya et al., 2020;Wang et al., 2021;Grythe et al., 2014), the net effect of different pathways may be subject to the uncertainties in the emission estimation and variation in controlling factors of the various pathways. Therefore, in order to better understand the role of halogen chemistry on tropospheric OH, we explore the

pathways by which halogen species influence OH, and how these pathways interact with each other and how they are influenced by different species-related processes in this study, based on current knowledge about halogen chemistry and marine emissions of halogen species. We carried out model simulations to quantify the contributions of different pathways by using a regional CTM (Community Multiscale Air Quality Modeling System, CMAQ) with process analysis (PA, including Integrated Process Rate, IPR, and Integrated Reaction Rate, IRR) over East Asia and Western Pacific during

summer. The controlling factors of strengths of the different pathways, mainly represented by different species-related processes, are analyzed based on PA and relevant sensitivity simulations, and their interactions are discussed. With higher spatial resolution than global models, we also explore the interaction of anthropogenic emissions with marine halogen emissions when discussing the controlling factors of strengths of the different pathways in iodine chemistry. The emission uncertainties are taken into consideration by running sensitivity simulations using the largest or the smallest emission rates

that have been used or reported in previous studies. The setup of the models and the estimations of marine emission and its extreme uncertainties of halogen species are described in Section 2. Results and discussions are in Section 3. Section 4 gives conclusions.

## 2. Methods

### 2.1 Model setup

We use the CMAQ model, driven by meteorological fields from Weather Research and Forecasting Model (WRF), to explore the impact of halogens on OH. For WRF (version 3.9.1), the domain has a horizontal resolution of 27km and the number of grids is 283×184. The vertical coordinates contain 39 sigma levels up to 50 hPa. The initial and boundary conditions are generated from the NCEP GDAS/FNL 0.25° analysis data. Analysis and observation nudging are applied. The data used for observation nudging are obtained from NCEP datasets *ds461.0* (for surface) and *ds361.1* (for upper layer). For

major physical parameterizations, the Rapid Radiative Transfer Model (RRTM) longwave radiation scheme, the Dudhia

shortwave radiation scheme, the WRF Single-Moment 3-class microphysics scheme, the Noah Land Surface Model, and the Grell-Freitas ensemble cumulus scheme are applied.

CMAQ (version 5.3.2) (Appel et al., 2021) has the same horizontal resolution as WRF, but with a slightly smaller domain. The vertical layers are the lowest 20 layers plus six of the remaining 19 layers of WRF. The chemical mechanism adopted here is CB6r3m released in CMAQv5.3, which is updated by adding halogen chemistry to CB6r3 mechanism based on the work of Sarwar and co-workers (Sarwar et al., 2019;Sarwar et al., 2015;Sarwar et al., 2014;Sarwar et al., 2012). Details of the gaseous reactions and heterogeneous reactions can be found in the recent work of Sarwar et al. (2019) and Sarwar et al. (2012) (see also Table S1). A Rosenbrock (ROS3) solver is used to solve the chemical reactions and the absolute and relative error tolerances are set to $10^{-9}$ ppm and $10^{-3}$, respectively. The initial and boundary conditions for CMAQ are extracted from a seasonal average hemispheric CMAQ output file that is obtained from the CMAS data warehouse (https://github.com/USEPA/CMAQ/blob/master/DOCS/Users_Guide/Tutorials/CMAQ_UG_tutorial_HCMAQ_IC_BC.md, last access: 6 October 2021). This hemispheric CMAQ used the same chemical mechanism as ours. The anthropogenic emissions are from MEIC (http://www. meicmodel.org/), while the emissions in the Guangdong province are replaced by local emissions that are based on a local emission inventory (Yin et al., 2015;Zheng et al., 2009) and processed by the Sparse Matrix Operator Kernel Emissions (SMOKE) processor. No halogen species are contained in the anthropogenic emissions. The terrestrial biogenic emissions are processed by MEGAN2.1 (Guenther et al., 2012). Other routine configuration set up of the model can be referred to Fan et al. (2021). Because OH concentration is highest in summer, the simulations of this study are for the month of July 2019, including an additional 10 day in June for spin-up following Li et al. (2020).

## 2.2 Marine emissions of halogen species

There are three main types of halogen species emitted from the ocean: SSA (Cl and Br), inorganic iodine ($I_2$ and HOI), and halocarbons including $CHBr_3$, $CH_2Br_2$, $CH_2BrCl$, $CHBr_2Cl$, $CHBrCl_2$, $CH_3I$, $CH_2ICl$, $CH_2IBr$, and $CH_2I_2$ (e.g., Sarwar et al., 2019;Carpenter et al., 2013;Ordóñez et al., 2012;Wang et al., 2019). The latest release version of CMAQ (v5.3) contains these emissions online.

The SSA emission in current CMAQ is updated by Gantt et al. (2015) on the top of the work of Kelly et al. (2010). The source function is based on the widely used source function developed by Gong (2003) which is an update of Monahan et al. (1986). Two main changes were implemented by Gantt et al. (2015). One is to add an SST-correction function to the source function because SST has large impacts on SSA flux (e.g., Barthel et al., 2019;Liu et al., 2021). The other is to change the shape factor of the source function (which determines the shape of the flux distribution) to emit more submicron SSA (see Fig. S1 of Gantt et al. (2015)). The SST-correction function is based on the work of Ovadnevaite et al. (2014) and is linear. This is different from another widely used observation-based SST-correction function developed by Jaeglé et al. (2011) which is a 3-order function of SST, but at high temperature (~30°C) their values are close (see eq. 2 of Gantt et al. (2015) and eq. 4 of Jaeglé et al. (2011)). In addition to these two main changes, surf-enhanced emission is also reduced by

narrowing the surf zone which was previously defined as 50 m to the coast and now reduced to 25 m as in the study of Gantt et al. (2015).

Inorganic iodine and halocarbons, as well as Br in SSA, are implemented as by Sarwar and co-workers (Sarwar et al., 2019). Inorganic iodine emissions are based on the work of Carpenter et al. (2013) which parameterized the emission of $I_2$ and HOI as functions of $O_3$ concentration, aqueous iodine concentration, and surface wind speed (see eqs. 19 and 20 in the SI of Carpenter et al. (2013)). Halocarbon emissions are calculated based on the work of Ordóñez et al. (2012) which directly related flux of halocarbons to chlorophyll-*a* (chl-*a*) concentration.

Current estimations of marine halogen emissions have large uncertainties. There are many different source functions of SSA, and the difference of the SSA flux calculated based on these source functions is very large (Grythe et al., 2014). The parameterizations of aqueous iodine have also different versions and differ largely (MacDonald et al., 2014;Sherwen et al., 2019;Chance et al., 2014). The halocarbon emissions are entirely empirical and have few physical bases. Therefore, it is necessary to consider the influence of the uncertainty in the emissions on final results. We design two simulation groups with

different emission rates, one high and one low. The high and low emission rates are taken from previously used estimations, similar to the work of Sekiya et al. (2020). The low emission rate of SSA is calculated using the source function in Gong (2003) directly, while the high emission rate using the source function modified by Gantt et al. (2015) because adding SST-correction function is somewhat more important than using different source functions (Barthel et al., 2019) and the source function of Gong (2003) or its modifications are the most widely used one. The parameterizations of $I_2$ and HOI emissions

are less variable and only that by Carpenter et al. (2013) is widely used. However, there are two widely used parameterizations of aqueous iodine with large difference. Therefore, the low emission rate of $I_2$ and HOI is calculated using low concentration of aqueous iodine, taken from MacDonald et al. (2014), while the high emission rate using high concentration, taken from Chance et al. (2014). The calculation of halocarbon emissions, which is based on the estimation of Ordóñez et al. (2012), is constrained by global annual flux (Sarwar et al., 2015); therefore, we increase or decrease

halocarbon emissions based on the ratios of global annual halocarbon fluxes reported by WMO (Engel et al., 2019) to that in Ordóñez et al. (2012). The scale factors are shown in Table S2. The chl-*a* data are obtained from the merged products of the GlobColour data set (http://globcolour.info, last access: 6 October 2021) that is developed, validated, and distributed by ACRI-ST, France.

The emissions of inorganic iodine are accompanied by the consumption of $O_3$ at the ocean surface. An enhanced $O_3$ dry

deposition by oceanic iodine is usually added (Luhar et al., 2018;Fairall et al., 2007;Luhar et al., 2017). In CMAQ, this $O_3$ deposition to ocean is based on the work of Chang et al. (2004), and uses the oceanic iodine concentration parametrization by MacDonald et al. (2014) (Sarwar et al., 2015). We use the aqueous iodine parameterizations consistent with that in the calculation of inorganic iodine emissions above.

To investigate the contribution from different species and pathways, we in total carried out more than eight simulation runs

other than the control run (BASE) in this study. The description of all the simulations and their differences are described in Table 1 (see also Tables S3) and the cross reference between cases and figures in this study is shown in Table S4.

**Table 1. Case design in this study.**

| Simulation case | Species or reactions[a] | Emission rate and ref |
|---|---|---|
| **BASE** | No halogen emissions in the domain | 0 |
| **BASE_phy** | As BASE but excluding[b] reaction $N_2O_5(g)+Cl(s)$, corresponding to SSA_phy below | 0 |
| **All_high** | SSA | High, from Gantt et al. (2015), $\approx$ Gong (2003) with SST correction from Ovadnevaite et al. (2014) |
| | $I_2$ and HOI | High, Carpenter et al. (2013) parameterization and Chance et al. (2014) aqueous iodine |
| | halocarbons | High, Ordóñez et al. (2012) parameterization and enhancement based on Engel et al. (2019) |
| **All_low** | SSA | Low, Gong (2003) |
| | $I_2$ and HOI | Low, Carpenter et al. (2013) parameterization and MacDonald et al. (2014) aqueous iodine |
| | halocarbons | Low, Ordóñez et al. (2012) parameterization and diminution based on Engel et al. (2019) |
| **SSA (SSA_Cl+Br)** | Only SSA | As in All_high |
| **SSA_Cl** | As SSA but excluding Br | As in All_high |
| **SSA_phy** | As SSA_Cl but excluding[b] the activation reaction $N_2O_5(g)+Cl(s)$ | As in All_high |
| **SSA_chemCl** | SSA_Cl−SSA_phy | -- |
| **SSA_chemBr** | SSA−SSA_Cl | -- |
| **InorgI** | Only $I_2$ and HOI | As in All_high |

| InorgI_chem | As InorgI but excluding enhanced O₃ dry deposition | Chang et al. (2004) & Sarwar et al. (2015) |
|---|---|---|
| O3depo | InorgI−InorgI_chem | -- |
| HaloC | Only halocarbons | As in All_high |

[a] All cases implement full halogen chemistry as in Sarwar et al. (2019) and Sarwar et al. (2012) except otherwise stated.

[b] The reaction is unchanged but the uptake coefficient of $N_2O_5$(g) and the yield of $ClNO_2$ are set to 0.

## 3. Results and discussions

### 3.1 Performance of the model

To evaluate the performance of our models, O₃, the key species for OH primary production, is compared between simulated and observed data over land (in China) and an island (Yonaguni) just east of Taiwan. The metrics for evaluation include the average observation (Obs_mean) and simulation (Sim_mean) values, root mean square error (RMSE), normalized mean bias (NMB), normalized mean error (NME), correlation coefficient ($r$), and index of agreement (IOA). The benchmarks are taken from the study of Emery et al. (2017). The statistical metrics of all stations is calculated and the average values are presented in Table 2. We evaluate stations in the three major polluted areas near the seas in mainland China, namely, the North China Plain (NCP), the Yangtze River Delta (YRD), and the Pearl River Delta (PRD) (Fig. S1a). For O₃ over the ocean, where is more relevant to this study, we obtain the measurements at the Yonaguni island (24.467°N, 123.011°E) to validate our simulation (data accessible at https://ebas.nilu.no/, last access: January 11, 2022) (Torseth et al., 2012). It can be seen that the O₃ concentrations are also reasonably simulated (Fig. S1b). In addition, adding the halogen emissions (especially with low emission rates) can noticeably lower the bias for the high ozone concentration (i.e., days before July 22) and improve the correlation between observation and simulations, which indicate the potential to improve the capability of ozone forecast at coastal stations by adding the marine-halogen emissions in the regional CTMs. Anyway, except NMB in the YRD, all these values meet the benchmarks (Emery et al., 2017), which shows that the model performance is comparable to those applications in different regions in China (Gao et al., 2020a;Li et al., 2022;Gao et al., 2022;Yao et al., 2020) and sufficient for our application.

**Table 2. Model performance metrics for 1-hr O₃ in mainland China and at Yonaguni island. The benchmarks are taken from Emery et al. (2017).**

| Region | Obs_mean (µg/m³) | Sim_mean (µg/m³) | RMSE (µg/m³) | NMB | NME | $r$ | IOA |
|---|---|---|---|---|---|---|---|
| NCP | 157.84 | 166.95 | 47.11 | 0.06 (< ±0.15) | 0.24 (< 0.25) | 0.61 (> 0.5) | 0.75 |
| YRD | 134.46 | 154.48 | 52.56 | 0.15 (< ±0.15) | **0.31** (< 0.25) | 0.59 (> 0.5) | 0.71 |

| | | | | | | | |
|---|---|---|---|---|---|---|---|
| PRD | 132.93 | 141.79 | 41.48 | 0. 06 (< ±0.15) | 0.23 (< 0.25) | 0.74 (> 0.5) | 0.83 |
| Whole region | 125.13 | 140.44 | 38.20 | 0.13 | 0.25 | 0.57 | 0.70 |
| Yonaguni | 39.20 | 41.82 | 14.14 | 0.07 (< ±0.15) | **0.26** (< 0.25) | 0.78 (> 0.5) | 0.84 |

Note. There is a threshold value of 40 ppbv for observations in the mainland China as recommended by Emery et al. (2017). For data at Yonaguni no threshold is applied because there is no significant diurnal cycle of $O_3$ concentration.

Figure S2 indicates a pretty good performance of the aerosol optical depth (AOD) stimulation that is important for the extinction effect of SSA as discussed in section 3.4.2.

For the relevant halogen species, although the in situ observational data over the marine area are limited, the model skills of marine halogens could generally be evaluated by the levels of BrO and IO due to their importance in halogen chemistry and the availability of the ship- and aircraft-based data and satellite remote sensing data (Li et al., 2020;Stone et al., 2018;Saiz-Lopez and von Glasow, 2012). Observations of BrO and IO are very rare around the world, especially in East Asia Seas. The available measurements of mean concentrations of BrO in Western Pacific show 1.0, 1.7 and <0.5 pptv in three flights

Koenig et al. (2017) and 0.69 pptv (Le Breton et al., 2017) during two related campaigns (CONTRAST and CAST). These values are generally smaller than measurements in the Atlantic Ocean (e.g., Read et al., 2008). In addition, according to the global model results (Zhu et al., 2019) and satellite remote sensing (e.g., http://www.doas-bremen.de/bro_from_gome.htm, last access: 4 June 2021), surface BrO concentrations have large annual variations in the Western Pacific, with the largest values in January and the smallest values in July. In a cruise in October from Japan to Australia, Großmann et al. (2013)

measured IO, showing that the daytime average of the IO concentration ranges from ~0.5 to ~1.5 pptv, with a typical daytime value ~1 pptv. Previous model results showed that surface IO in the Western Pacific has a significant seasonal variation, peaks in summer (Huang et al., 2020) and the difference between July and October is about 0.2–0.4 pptv according to their Figs. 3m,p. Therefore, it is expected that our simulation values (in July) will be slightly larger than the values reported by Großmann et al. (2013). Moreover, since modelled IO also decreases with height in the lower troposphere (see

Fig. 2 of Huang et al. (2020)), the surface IO is also expected to be slightly larger than the boundary-layer average of IO.

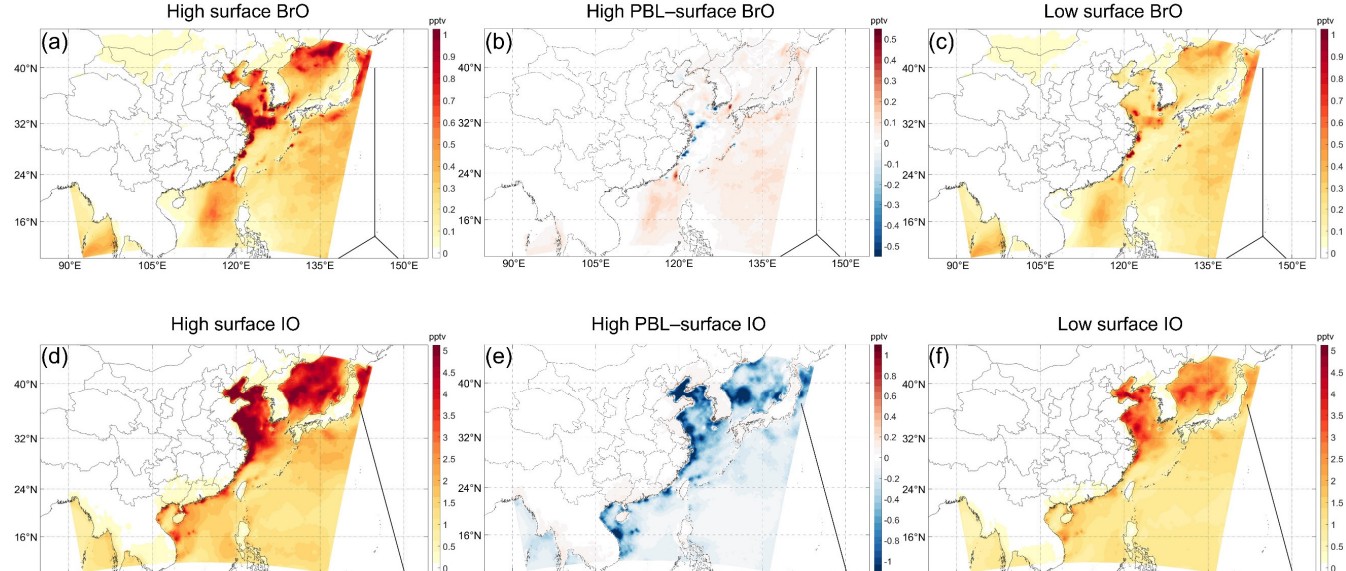

**Fig. 1. Daytime (local time 8:00–16:00) average of (a) surface BrO, (b) difference between planetary boundary layer (PBL) and surface BrO, for high emission rate, and (c) surface BrO for low emission rate. (d)–(f) are for IO. The black lines indicate roughly the trajectories of flights or cruise in previous studies reporting relevant measurements (see Table 3).**


**Table 3. Comparison of BrO and IO in the Philippine Sea in our simulations (surface layer) with observations and simulations reported in other studies.**

| | Mean/max | | | Platform |
|---|---|---|---|---|
| | Simulation, low emission rate | Simulation, high emission rate | Observation or model | |
| BrO (pptv) | 0.2/0.9 Jul | 0.25/1.2 Jul | ~1/2.9[a,1] Jan & Feb | Flights around Guam (line in Figs. 1a–c) < ~500m |
| | | | 0.69/1.71[2] Jan | |
| | | | ~1/>2[3] Jan | ~0.3/>0.6[3] Jul | GEOS-Chem surface layer |

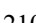

| | | | ~1/~1.5[4] Oct | Cruise from Japan to Australia (line in Figs. 1d–f) PBL |
|---|---|---|---|---|
| IO (pptv) | 1.0/1.8 Jul | 1.4/2.5 Jul | | |
| | | | Average Jul > Oct by ~0.2–0.4[5] | CMAQ surface layer |

[a] only data at altitudes below 500 m.

[1] Koenig et al. (2017), [2] Le Breton et al. (2017), [3] Zhu et al. (2019), [4] Großmann et al. (2013), [5] Huang et al. (2020).

Figure 1 shows the daytime (local time 8:00-16:00) average BrO and IO simulated in our studies. Due to the lack of observation data in the coastal seas for comparison, we only discuss the results in the Philippine Sea (i.e., the open ocean east to the line connecting the Philippines, Taiwan, and Japan). In this sea, the concentrations of BrO and IO are generally lower than nearshore areas. The maximum mean values of the daytime BrO and IO are 1.2 (0.9) and 2.5 (1.8) for high (low) emissions; for the average over all these grids, the daytime BrO is about 0.25 (0.2) pptv, while IO about 1.4 (1.0) pptv for

high (low) emission rates. For boundary-layer average, the values for IO are lower than surface values by ~0.08 pptv for grid average and by ~0.4 pptv for grid maximum in the Philippine Sea in All_high case (Fig. 1d) (due to the storage limitation, we did not output upper-layer results in All_low case). Different from IO, BrO does not decrease with height in the lower ~500m in the study of Huang et al. (2020), and our simulations also show that boundary-layer average of BrO is slightly larger than surface values by ~0.05 pptv (Fig. 1b).

Table 3 lists the comparison of the available measurements and global model results in the area and our model results. It can be seen that our model results generally agree well with measurements and other model results. It should be emphasized that the comparison is only indirect and there is a lack of data for even indirect comparison in the nearshore areas where the IO concentration is the largest. Since the inorganic iodine emission is closely related to $O_3$ concentration which is high in the nearshore areas due to the outflow from the continent, the higher concentration of IO is reasonable, and in other regions,

observations also support a very high concentration of IO in nearshore areas (Saiz-Lopez and von Glasow, 2012), but nevertheless, relevant observations are expected for a better validation.

## 3.2 The changes of OH production rate ($P_{OH}$) and concentration

Figure 2 illustrates the halogen-induced changes of $P_{OH}$ and OH concentration in All_high (all halogen species high emission rates) and All_low (all halogen species low emission rates) cases. $\Delta P_{OH}$ and $\Delta OH$ with both high and low emission rates all

have similar spatial distributions but with different magnitudes. The most significant changes of $P_{OH}$ and OH appear in the marine atmosphere (Fig. 2). The impacts are very complicated, with negative $\Delta P_{OH}$ and $\Delta OH$ in the middle area of the ocean while positive in the north and south parts of the ocean in the domain, but the area with negative impacts is larger than that

with positive impacts. The decreases of OH can reach ~13% and ~8% ($\Delta P_{OH}$ ~15% and ~10%) in the Philippine Sea and the increase can reach ~11% and ~9% ($\Delta P_{OH}$ ~9% and ~7%) in the Bohai Sea, with high and low emission rate, respectively.

This is in line with previous studies that generally showed decrease of globally-averaged OH but certain increase in some regions due to halogen chemistry (e.g., Sherwen et al., 2016;Stone et al., 2018). More specifically, in the East Asia seas, the studies of Stone et al. (2018), Wang et al. (2019) and Sherwen et al. (2016) generally showed a slight decrease (<~5%) of annual-averaged surface OH while Stone et al. (2018) also showed a slight increase in some regions. For studies in July, the study of Li et al. (2019) showed a decrease of monthly averaged surface OH in the Atlantic Ocean near to Europe but an increase in the Mediterranean Sea and the Baltic Sea. The decrease in the Atlantic Ocean can reach ~20% in the middle latitude. In the Indian Ocean, Mahajan et al. (2021) showed a slight decrease (<5%) of monthly averaged surface OH near the Indian subcontinent while increasing (<10%) near the equator, and the area with decreased OH is larger than that with increased OH in their model domain. In the coastal areas the absolute changes of $P_{OH}$ and OH can be comparable to or even larger than that over the ocean, but the relative values are relatively small due to the large absolute value over land (Figs. 2b, d, f, h). The largest decreases of monthly $P_{OH}$ and OH can reach ~3–5% and ~4–6%, respectively (Figs. 2b,f).

Generally speaking, our results are comparable to previous studies, showing overall negative halogen-induced $\Delta OH$ but with complicated spatial distribution of negative and positive $\Delta OH$ (and $\Delta P_{OH}$), especially in nearshore area. Previous studies have qualitatively and partially explained the reasons why halogens have such a complicated impact on OH, as the two pathways by which halogens influence OH (i.e., enhanced $HO_2$ conversion by XO and $O_3$ consumption by X atoms) have opposite impacts on OH (e.g., Stone et al., 2018). However, the complicated spatial distribution of negative and positive $\Delta OH$ indicates a complicated interaction of the pathways. Furthermore, it is unclear whether there are other important pathways by which halogens influence OH. Therefore, in order to better understand the impacts of halogens on OH, and more specifically, to understand why halogens increase OH in certain regions (especially nearshore area) but decrease in other regions, we need to find out all possible important pathways, and to further analyze the controlling factors of the strengths of the pathways.

In the following, we will further analyze the causes of such a complicated distribution. Since the spatial distributions of relative $\Delta P_{OH}$ and $\Delta OH$ are very similar despite the small difference in magnitudes, and the OH chemistry is generally discussed in terms of $P_{OH}$ in the literature (e.g., Tan et al., 2019;Hofzumahaus et al., 2009;Whalley et al., 2021) and we can directly separate different pathways by which influence $P_{OH}$, we will focus on $P_{OH}$ in the following. In addition, because the patterns of $\Delta P_{OH}$ and $\Delta OH$ (Fig. 2), as well as the IRR results (Figs. 4 and S3), are quite similar in the All_high and All_low cases (Fig. 2), we will mainly focus on cases with high emission rates.

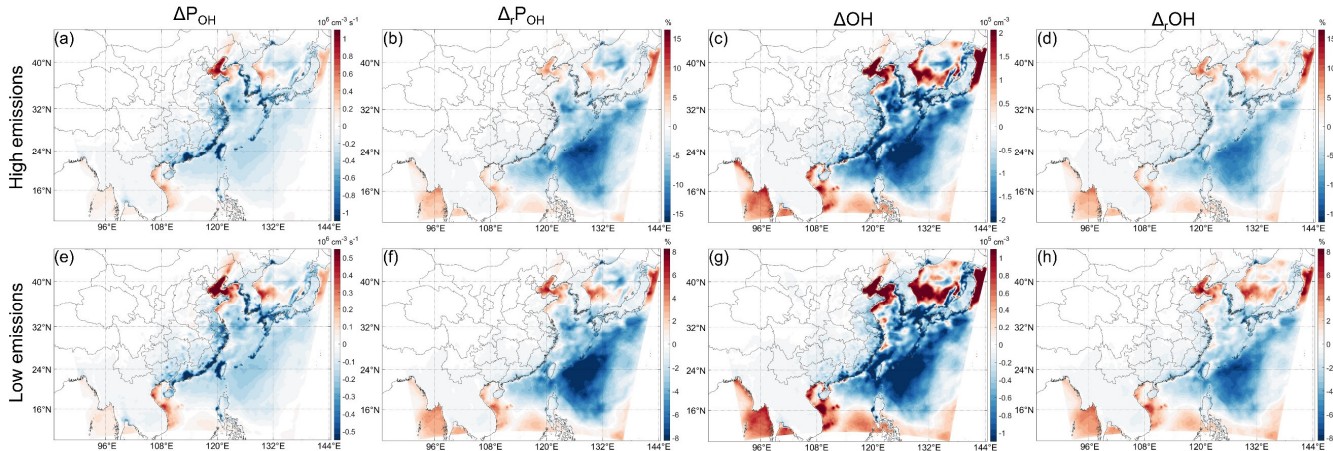

**Fig. 2. Change or relative change compared to BASE case of monthly averaged surface-layer P$_{OH}$ and OH in All_high case (first row) and All_low case (second row). The subscript "r" denotes "relative". Note the different scales in the All_high and All_low cases, the latter exactly a half to the former.**

### 3.3 Quantification of different pathways' contributions

As mentioned above, there is complexity in the cause of the $\Delta$P$_{OH}$. In this section, IRR is used to unravel important chemical reactions in changing P$_{OH}$. The main sources of OH in the CB6 mechanism of CMAQ model include primary sources and secondary sources. Primary sources include the photolysis of O$_3$ (through the reaction O($^1$D)+H$_2$O, which will not always be explicitly stated in the following), HONO, and H$_2$O$_2$, and ozonolysis of some alkenes. The secondary source is mainly the reactions HO$_2$+Y (Y=NO, O$_3$, etc.). With halogen chemistry, an additional source, HOX photolysis, needs to be considered as HOI can be directly emitted and can be very rapidly cycled. The changes of P$_{OH}$ due to the change of all these sources (denoted as $\Delta$P$_{OH\_XX}$ in the following where XX is clear from the context) based on IRR analysis are quantified. According to the IRR results, we only focus on the main three changes (photolysis of O$_3$ and HOX, and the reaction HO$_2$+Y) (Fig. 3a–c). Since the changes of other sources are ignorable (Fig. 3d), we do not show them individually. We denote the halogen-induced change of these sources as pathways by which halogens influence OH and therefore there are three main pathways through which marine-emitted halogens influence P$_{OH}$, i.e., P$_{OH\_O1D}$, P$_{OH\_HO2}$, and P$_{OH\_HOX}$.

In line with previous studies, the results show the change of O$_3$ and the addition of HOX are the two most important pathways by which halogens influence P$_{OH}$ (Stone et al., 2018). $\Delta$P$_{OH}$ caused by the change of O$_3$ and HOX photolysis (denoted as $\Delta$P$_{OH\_O1D}$ and $\Delta$P$_{OH\_HOX}$ respectively) are very large in the north part of the ocean in the domain, especially in the Bohai sea and the Yellow sea, which is probably a result of the higher concentration of related species such as O$_3$ that is commonly reported at high concentration in the midlatitude in summer (e.g., Gao et al., 2020b;Lu et al., 2019b;Hu et al., 2017). $\Delta$P$_{OH\_HOX}$ (Fig. 3f) can reach $4\times10^6$ cm$^{-3}$ s$^{-1}$ (~0.6 ppbv h$^{-1}$) for the whole-day average and $1\times10^7$ cm$^{-3}$ s$^{-1}$ (~1.5 ppbv h$^{-1}$) for the daytime average. Our results show that HOX is an important source of OH over the ocean (may be compared to urban-area HONO), but it was generally ignored in the previous HO$x$ budget studies (e.g., Tan et al., 2019;Hofzumahaus et

al., 2009;Whalley et al., 2021). Therefore, our results indicate the necessity to measure HOX in HO$x$ budget studies under the potential influence of marine atmosphere.

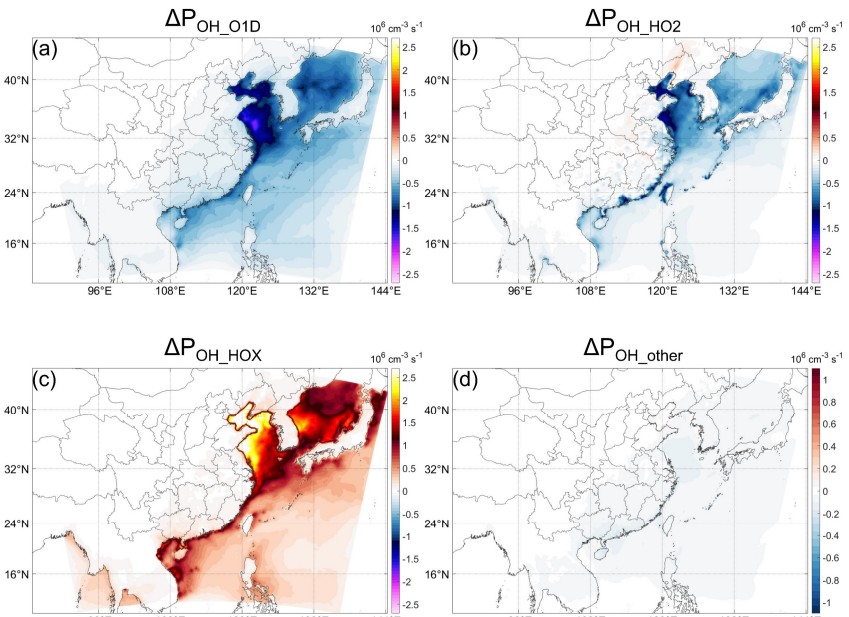

**Fig. 3. Decomposition of Fig. 2a for different pathways: change of P$_{OH}$ (All_high–BASE case) caused by the changes of (a) O($^1$D) (ΔP$_{OH\_O1D}$), (b) HO$_2$+Y (ΔP$_{OH\_HO2}$, Y=NO, O$_3$, etc.), and (c) HOX+hv (ΔP$_{OH\_HOX}$), and (d) other pathways (=Fig. 2a minus sum of (a)–(c)). The red and blue color scales are the same in (a)–(d).**

In addition to ΔP$_{OH\_O1D}$ and ΔP$_{OH\_HOX}$, ΔP$_{OH\_HO2}$ is also very important to ΔP$_{OH}$ as shown in Fig. 3b. If we considered only ΔP$_{OH\_O1D}$ and ΔP$_{OH\_HOX}$, only a relatively small area close to the Taiwan island would show negative ΔP$_{OH}$ and the general impacts of halogens on OH would be positive (compare Figs. 2a and S4a).

As mentioned above, the production rate of OH from HOX is very large. However, this large production rate is canceled by the large decrease of ΔP$_{OH\_O1D}$ and ΔP$_{OH\_HO2}$ (Figs. 3a,e), resulting in the relatively small net ΔP$_{OH}$ compared to ΔP$_{OH\_O1D}$ and ΔP$_{OH\_HOX}$ (and even ΔP$_{OH\_HO2}$ for many regions) over the ocean but still significant along the coastal areas (Fig. 3). It can be seen that the cancel-out effect of the three pathways with different signs results in the complicated spatial distribution of ΔP$_{OH}$, making ΔP$_{OH}$ positive in the areas with larger ΔP$_{OH\_HOX}$ and negative otherwise. From these three pathways themselves, however, it is difficult to explain under what conditions ΔP$_{OH\_HOX}$ will be stronger than the other two pathways, and therefore difficult to explain why ΔP$_{OH}$ is generally positive in the nearshore areas while negative in the open ocean. Then we need to further analyze the details of the processes influencing the strengths of these three pathways.

## 3.4 Factors influencing the strengths of the three pathways

### 3.4.1 Overview of the contributions from different marine-emitted species

There are several factors that can change the strengths of the three main sources of OH (Fig. S5). Some of these factors are independent, some are interrelated. The independent factors include decrease of $O_3$ photolysis rate ($J(O^1D)$) and $O_3$ concentration by SSA-induced light extinction, and enhancement of $O_3$ deposition by oceanic iodine. The interrelated factors are generally closely related to halogen chemistry of which the most important reactions are the below three that for all three halogen elements and the fourth for Cl only (Saiz-Lopez and von Glasow, 2012;Simpson et al., 2015):

$$X + O_3 \rightarrow XO + O_2 \quad\quad (R1)$$

$$XO + HO_2 \rightarrow HOX + O_2 \quad\quad (R2)$$

$$HOX + h\nu \rightarrow X + OH \quad\quad (R3)$$

$$Cl + VOCs \xrightarrow{O_2} RO_2 + HCl \quad\quad (R4)$$

For convenience, we also list the two reactions producing OH that are relevant to R1–R3 and have been mentioned above:

$$O_3 + h\nu \rightarrow O^1D + O_2 \quad\quad (R5a)$$

$$O(^1D) + H_2O \rightarrow 2OH \quad\quad (R5b)$$

$$HO_2 + NO \rightarrow OH + NO_2 \quad\quad (R6a)$$

$$HO_2 + O_3 \rightarrow OH + 2O_2 \quad\quad (R6b)$$

Since these factors just mentioned above are generally species-related, we separately modeled the impacts of different

halogen species in addition to the case with all emissions (All_high) (Table 1). The results are shown in Fig. 4. It can be seen that the most significant contributors to the three pathways are inorganic iodine (Fig. 4e–g). However, the three pathways cancel out each other to a large extent and the resultant $\Delta P_{OH}$ is relatively small. Nevertheless, the impact of inorganic iodine is more pronounced than that of all species together (Figs. 2a and 4e). The contribution of SSA to $\Delta P_{OH}$ is notable, comparable to that of inorganic iodine in most regions. There is positive contribution of SSA to $\Delta P_{OH}$ in the Bohai Sea and

surroundings, while in other regions the contribution is negative. The negative contribution again neutralizes the positive contribution of inorganic iodine, resulting in the more negative $\Delta P_{OH}$ in All_high case (Fig. 2a) than in InorgI case (Fig. 4e). The contribution of halocarbons is relatively small and restricted to a small area near to the China coastline. In addition, the interactions between these three types of emitted species (Fig.4m) have very similar impacts with halocarbons (Fig. 4i) but with opposing sign. Since we only focus on major contributions of different halogen species to $\Delta P_{OH}$, we will not go into the

details about the rest of the interactions of the three types of halogen emissions, and therefore we also do not discuss the influences of halocarbons in the following as they roughly cancel out the effects of the interactions. It should be noted, however, this does not imply that the interactions are caused by halocarbons.

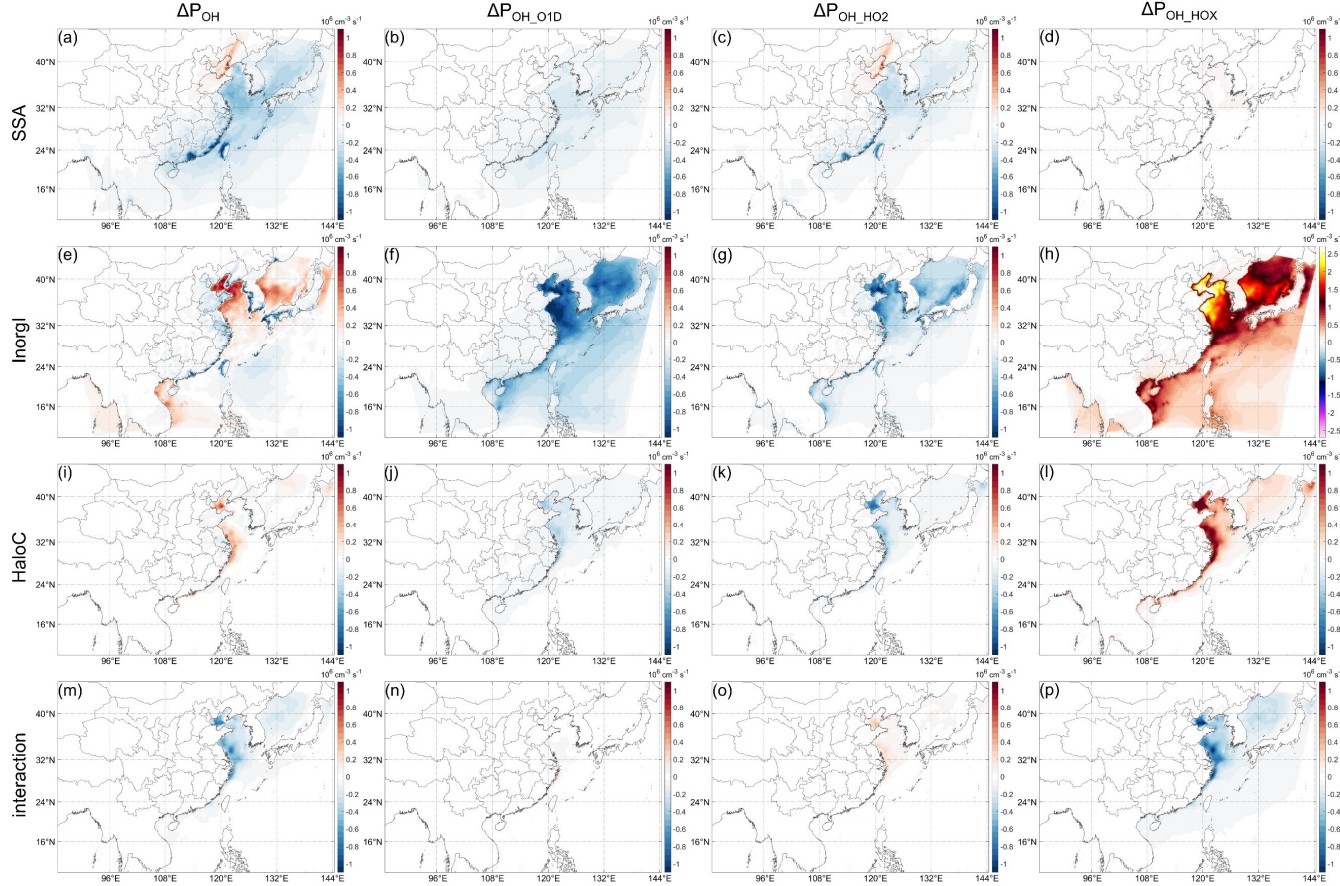

**Fig. 4. Decomposition of Fig. 3 for different halogen species. (a)–(d), (e)–(h), and (i)–(l) are results for SSA, InorgI, and HaloC case, respectively. (m)–(p) are the residue between All_high and SSA+InorgI+HaloC, representing the interactions of different halogen species. The red and blue color scales are the same in (a)–(p).**

### 3.4.2 Physical and chemical contributions of SSA emission

Regarding SSA, whose components are mainly Cl⁻ ions and inert non-volatile cations (NVCs, including Na⁺, K⁺, Ca²⁺, and Mg²⁺), with minor contribution from sulfate and Br⁻ ions, these emissions (see Fig. S6c,d for the emission rates) could influence OH through both actinic flux and chemical effects of Cl and Br. The SSA impact on OH (Fig. 4a−d SSA) is further decomposed into the impact of the extinction effect of SSA, Cl chemistry, and Br chemistry (Fig. 5). Figure 5 indicates that the most important factor that determines the negative impact of SSA on $P_{OH}$ (Fig. 4a) is its extinction effect (Fig. 5a–d). The decrease of actinic flux caused by the extinction effect of SSA can decrease the photolysis rate constant of $O_3$ ($J(O^1D)$) and the $O_3$ concentration (probably through influencing $J(NO_2)$, see e.g., Gao et al. (2020a)) at the same time (Fig. S6a,b), which will in turn decrease the OH production from $O_3$ photolysis (R5). The negative impacts of Br chemistry are very small compared to this extinction effect. In addition to the overall large impacts, the importance of the SSA extinction effect is also embodied in its impact on the continental atmosphere. As shown in Figs 4a,e,i the $\Delta P_{OH}$ over land induced by SSA is the

most significant among the three halogen emissions, and here we know that the relatively large decrease of $P_{OH}$ in southern China is caused by the extinction effect of SSA. The decrease of monthly $P_{OH}$ caused by SSA can reach ~3% (Fig. 6a)

(hourly $\Delta_r P_{OH}$ up to 30% in the daytime). Therefore, even without halogen chemistry, adding SSA emissions in CTM studies may be important for atmospheric chemistry.

Another important factor that influences $P_{OH}$ is the Cl chemistry (Figs. 5e−h and 6b). Similar to previous studies, Cl chemistry has positive impacts on $\Delta P_{OH}$ because Cl can oxidize VOCs, which can both come from anthropogenic and oceanic source (Yu and Li, 2021), efficiently and produce $RO_2$ radicals (R4) as shown in Fig. 5e (Li et al., 2020;Wang et al.,

2020;Simpson et al., 2015). As such, the change of $P_{OH}$ by Cl chemistry (Fig. 5e) is mostly through the change of OH production from $HO_2$ (Fig. 5g). The impacts of Cl chemistry are most significant in the Bohai Sea and surroundings. As shown in Fig. S7, in these areas, the concentration of $ClNO_2$ (the key species for the activation of SSA Cl) are higher than other regions (Fig. S7a) and the Cl reactivity ($k_{Cl}$, $=\sum k_{Cl+VOC} \times [VOC]$) is very high, resulting in the larger impact of the Cl chemistry. $ClNO_2$ is a product of $N_2O_5$ with particulate Cl, and $N_2O_5$ is a product of $NO_2$ and $NO_3$ radical (e.g., Yu et al.,

2020). Therefore, the larger impacts of Cl chemistry in the Bohai Sea and surroundings probably reflects the influence of higher NO$x$ in the area. The impact of Br chemistry on $P_{OH}$ is quite small in general (Figs. 5i–l), and we will not discuss it further (see more discussion about Br chemistry in section 3.5). Nevertheless, the results from Br chemistry emphasize the importance of pathway $P_{OH\_HO2}$ in interpreting the roles of halogens in influencing HO$x$ cycling: When considering the influence of halogens on HO$x$, it is believed that XO will shift the HO$x$ balance to OH in general (e.g., Li et al., 2020;Stone

et al., 2018;Saiz-Lopez and von Glasow, 2012). But these results are derived without considering the indirect influence of halogens on R6 (i.e., pathway 2, $P_{OH\_HO2}$, in our study). In our study, when the inhibition of halogen chemistry on the $HO_2$ conversion to OH through $HO_2$+Y is considered (R6), only IO can uniformly enhance the $HO_2$ conversion and BrO cannot in some areas (compare Figs. 5k and 5l, or see Fig. S4b) which is probably because the lower reactivity of BrO with $HO_2$ (see also discussion in section 3.4.2 of Whalley et al. (2010)).

In regards to the three main pathways discussed in section 3.3, the physical contribution of SSA is achieved not only through the decrease of the photolysis of $O_3$ ($J(O^1D)$ and the $O_3$ concentration) (Fig. 5b), but also through the decrease of $HO_2$ conversion to OH (Fig. 5c) which is probably feedback induced by the decrease of $O_3$ photolysis because $HO_2$ production is little influenced by photolysis change. In contrast, the chemical contribution of SSA Cl is achieved through the increase of $RO_2$ that can rapidly react with NO to form $HO_2$ (Seinfeld and Pandis, 2016), and therefore the second pathway, OH from

$HO_2$, is more prominent (Fig. 5g), while the increase of $O_3$ concentration is of minor importance (Fig. 5f).

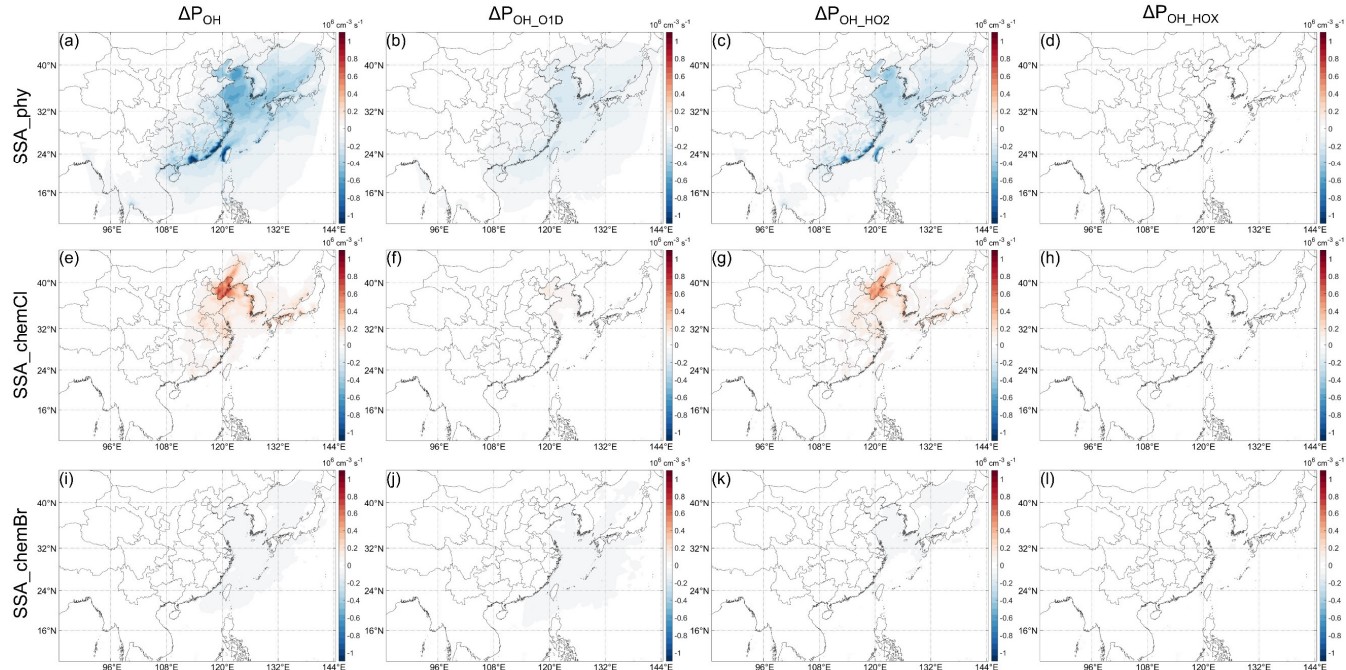

**Fig. 5.** Decomposition of Fig. 4a–d (SSA): the changes of $P_{OH}$, $P_{OH\_O1D}$, $\Delta P_{OH\_HO2}$, and $\Delta P_{OH\_HOX}$ caused by (a)–(d) the extinction effect of SSA, denoted as SSA_phy, (e)–(h) Cl chemistry (only the activation of Cl through ClNO₂), and (i)–(l) Br chemistry.

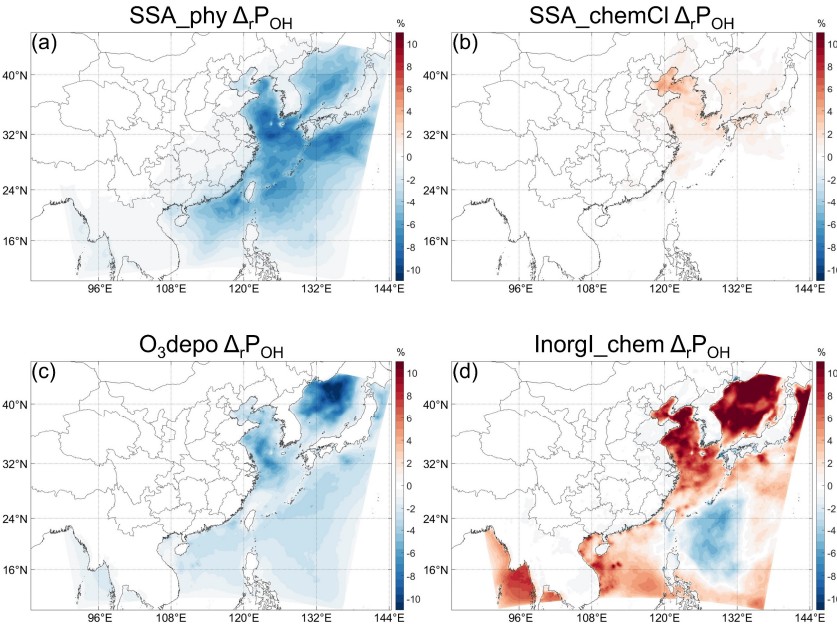

**Fig. 6.** Relative change of $P_{OH}$ compared to BASE, caused by (a) SSA extinction effect, (b) SSA Cl chemistry, (c) enhanced O₃ deposition by aqueous iodine and (d) atmospheric inorganic iodine chemistry.

### 3.4.3 Physical and chemical contributions of inorganic iodine species, and the interactions between $O_3$ and iodine chemistry

Regarding the contributions from inorganic iodine species on $\Delta P_{OH}$, the indirect effects from the enhanced $O_3$ deposition by iodine ion in the ocean (e.g., Pound et al., 2020) and atmospheric iodine chemistry (including the direct emission of HOI) should be considered. The change of $P_{OH}$ caused by the enhanced $O_3$ deposition is shown in Fig. 7a. This $O_3$-deposition-induced decrease of $P_{OH}$ is most significant in the Bohai and Yellow Sea, where it can reach ~$0.4\times10^6$ cm$^{-3}$ s$^{-1}$, corresponding to ~6% (hourly $\Delta_r P_{OH}$ can reach >30% in the daytime) in the Yellow Sea relative to $P_{OH}$ in the BASE case (Fig. 6c). The larger absolute decrease of $P_{OH}$ (Fig. 7a) is probably caused by the higher deposition of $O_3$ in the Bohai and Yellow Sea because of the higher $O_3$ concentration there as mentioned above (see also Fig. 8). For the relative change, the decrease of $P_{OH}$ is most significant in the Sea of Japan, where the relative decrease can reach more than 10% (hourly $\Delta_r P_{OH}$ can reach >45% in the daytime) (Fig. 6c). By decomposing to different pathways, the decrease of $P_{OH}$ induced by $O_3$ deposition are caused by the decrease of $O_3$ ($P_{OH\_O1D}$, Fig. 7b), and by the decrease of HOX photolysis ($P_{OH\_HOX}$, Fig. 7d) which probably results from the slower cycling of HOI through reactions R1–R3 due to the lower $O_3$ concentration.

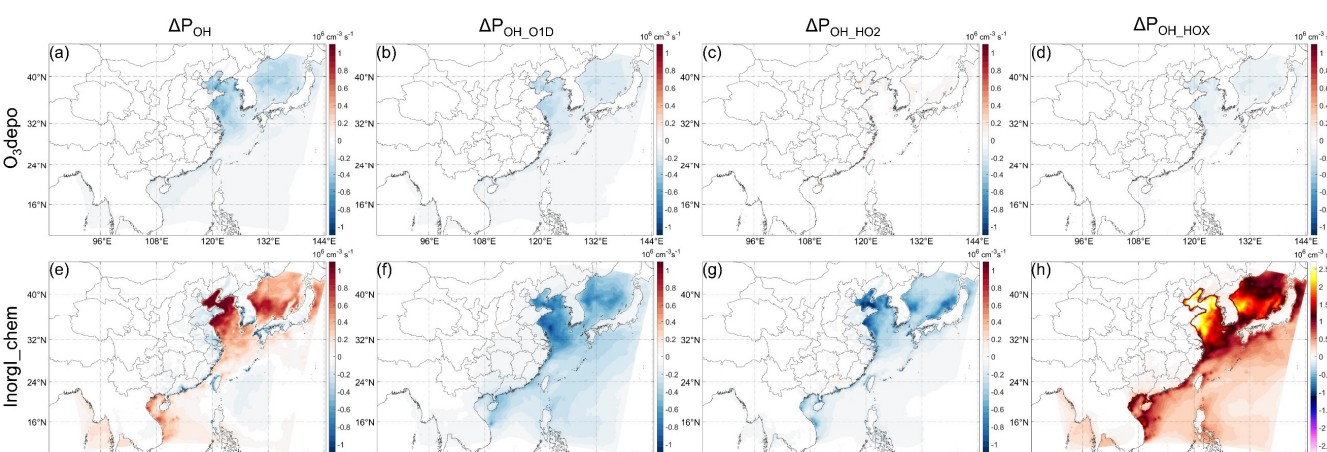

**Fig. 7. Decomposition of Fig. 4e–h (InorgI): the changes of net OH production rate caused by (a)–(d) the enhanced $O_3$ deposition by oceanic iodine ions and (e)–(h) atmospheric iodine chemistry (including direct HOI emission). The red and blue color scales are the same in (a)–(h).**

Regarding the contributions from the atmospheric iodine chemistry on $\Delta P_{OH}$, it has general positive impact on $P_{OH}$ except in a limited area (Figs. 6d and 7e). More specifically, the decrease of $\Delta P_{OH}$ is relatively significant in the Philippine Sea (up to ~8% of BASE case). The positive $\Delta P_{OH}$ in the Bohai and Yellow Sea can reach more than 10% of BASE case, and the relative increase can reach more than 15% in the Sea of Japan. The changes of $P_{OH}$ caused by the atmospheric iodine chemistry from different pathways are shown in Fig. 7e−h. It can be seen in Fig. 7 that the contributions of three pathways are all significantly influenced by iodine chemistry. Different from Cl chemistry, iodine chemistry can both increase $P_{OH}$ via R3 (Fig. 7h) and decrease $P_{OH}$ via R1−R2 (Figs. 7f−g).

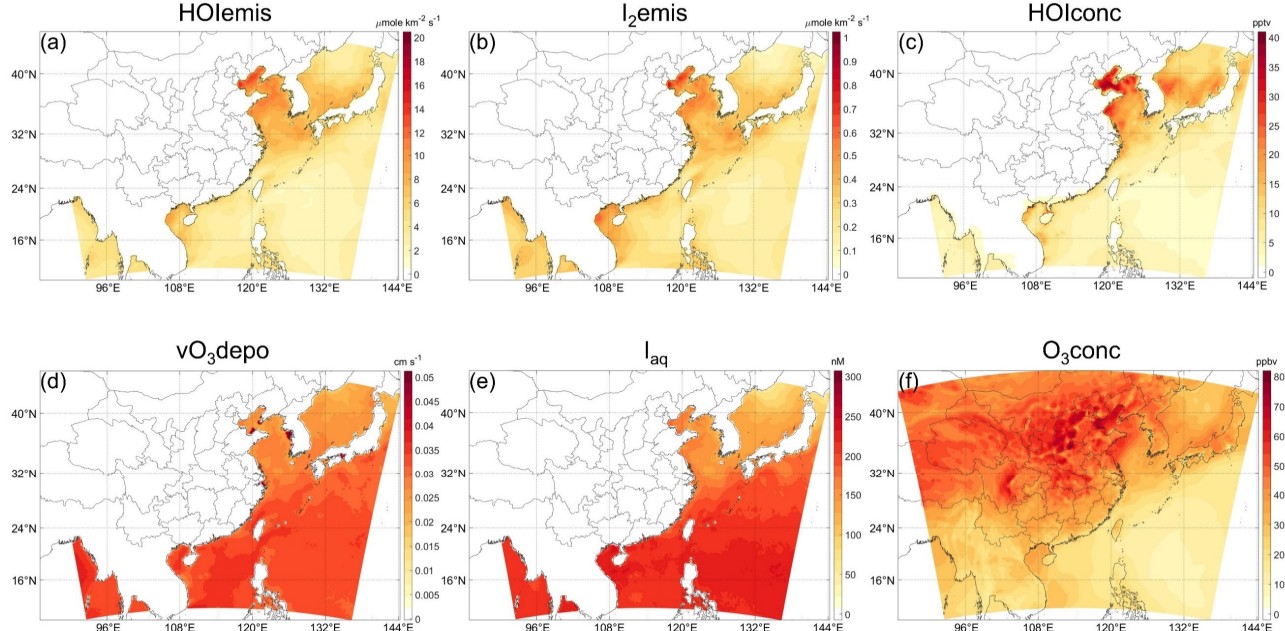

**Fig. 8. Monthly average of parameters related to emission of inorganic emission in All_high case. (a) HOI emission rate, (b) I₂ emission rate, (c) HOI concentration, (d) O₃ deposition velocity over the ocean, (e) seawater iodine ion concentration, and (f) O₃ concentration. Note the different scales of (a) and (b). (d) is comparable to the values in the study of Pound et al. (2020), an annual average O₃ deposition velocity about 0.03 cm s⁻¹ and in summer the deposition velocity is close to annual average in the Western Pacific, see their Figs. 2b and 3c.**

The negative change of $P_{OH}$ caused by atmospheric iodine chemistry in the Philippines Sea shown in Fig. 6d is interesting and to our knowledge there is no study previously published detailing the cause of the both positive and negative impacts of iodine chemistry (e.g., Stone et al., 2018). To investigate the factors driving the spatial variability of iodine-induced OH production, the monthly averaged values of the parameters related to emission of inorganic emission in All_high case are plotted in Fig. 8. As expected, the pattern of $\Delta P_{OH}$ does not coincide with the patterns of seawater iodine concentration (Fig. 8e) and the ozone deposition velocity (Fig. 8d), which decrease with latitude due to the increase of temperature. The location of the area with negative $\Delta P_{OH}$ roughly coincides with the area with lower emission rates of HOI and I₂ (Fig. 8a,b), and concentrations of HOI (Fig. 8c) and O₃ concentration (Fig. 8f). The positive change of $P_{OH}$ could be directly attributable to the effect of iodine chemical cycling, but what are the causes of the negative change of $P_{OH}$? To figure out the causes, we conduct sensitivity simulations to analyze the contribution of I₂/HOI emissions and O₃ concentration on the negative $\Delta P_{OH}$.

First, we find that the emission of HOI as a primary OH source is not the cause. The PA results show that the emission rate of HOI is much smaller than that of HOI cycling rate (Fig.S8a,b). Moreover, the change of $P_{OH\_HOX}$ by replacing direct HOI emission with an equivalent amount of I₂ using the emission scaling technique developed by Murphy et al. (2021) is relatively ignorable (Fig. S8c). Even without direct HOI emission (i.e., replaced by I₂) there can be also negative $\Delta P_{OH}$ in the area (Fig. S8d). Second, we show that the lower emission rate of inorganic iodine (emitted as HOI or I₂) in the Philippine Sea

is probably not the cause of the negative $\Delta P_{OH}$ there by two sensitivity runs (addressing the impacts of the absolute value and the spatial distribution of the emissions, respectively). When reducing $HOI/I_2$ emission rates by 50% uniformly, the area of

negative $\Delta P_{OH}$ does not increase (Fig. S9a,b), indicating that fewer available iodine atoms does not necessarily enlarge the area of negative $\Delta P_{OH}$ in the Philippines Sea. Furthermore, when we set $HOI/I_2$ emission rates in the whole domain to constants ($6.86=5000/27^2$ μmole km$^{-2}$ s$^{-1}$ for HOI, and 1/20 of HOI for $I_2$, see Fig. 8a,b for comparison) which are between the emission rates in the Philippine Sea and that in the Bohai Sea, the pattern of $\Delta P_{OH}$ is very similar to that in the InorgI_chem case (Figs. S9e and 7e). This similarity strongly suggests that the distribution of inorganic iodine emission is

also not important to determine the positive-or-negative pattern of iodine-chemistry-induced $\Delta P_{OH}$.

Is it the complex interaction between the marine iodine chemistry and the continental atmospheric pollution that leads to the special negative $\Delta P_{OH}$ in the Philippine Sea? Third, to investigate whether this is true, we conduct simulations increasing $O_3$ concentration downwind of the Philippines (Fig. S10a) by increasing NO$x$ and VOCs emission rates in the Philippines by a factor of five. The iodine-chemistry-induced change of $P_{OH}$ is negative in most of the Philippines Sea but positive near to the

land (Fig. S10c). This characteristic results from the different distributions of $\Delta P_{OH\_O1D}$ (Fig. S10d) and $\Delta P_{OH\_HOX}$ (Fig. S10e) along the $O_3$ plume: the former relatively evenly distributed while the latter weakening gradually along the plume. The gradual decrease of $\Delta P_{OH\_HOX}$ (via R2) along the plume path is similar to the $O_3$ distribution (Fig. S10a), because the HOI cycle is essentially local due to the very high cycling rate, leading to the local impact of $P_{OH\_HOX}$. In contrast, $\Delta P_{OH\_O1D}$ caused by the $O_3$ consumption via R1 does not gradually weaken along the plume as $O_3$ concentration and $\Delta P_{OH\_HOX}$ do,

which is because the decrease of $P_{OH\_O1D}$ (via R1) is not only controlled by the local $O_3$ consumption but also the upwind $O_3$ depletion along the plume. To confirm the impact of the "upwind $O_3$-depletion" by iodine (R1) in the Philippine Sea, the ratio of total $O_3$ concentration decrease to local $O_3$ loss rate is shown in Fig. 9c. Indeed, the ratio has maxima in the Philippine Sea where the iodine-chemistry-induced $\Delta P_{OH}$ are negative (Fig. 7e), indicating an "excessive" decrease of $O_3$ compared to its local consumption in the area (Fig. 9), which can only be explained by the gradual enhancement of "upwind

$O_3$-depletion" by iodine along the plume paths (in InorgI_chem case, compared to BASE case, iodine atom is the only significant consumer of $O_3$).

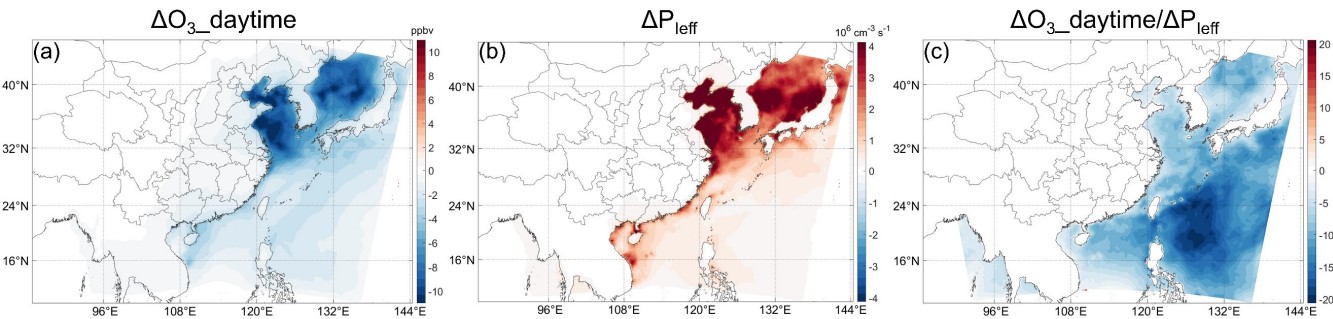

**Fig. 9. Iodine chemistry induced change of (a) daytime (LT 8:00–16:00) $O_3$ concentration and (b) production rate of effective iodine atom (iodine atom that can consume $O_3$, including all reactions that can produce iodine atom except IO photolysis which**
**will produce O and reactions which produce $NO_2$), $\approx O_3$ loss rate by local iodine chemistry. (c) The ratio between (a) and (b), only**

for qualitative illustration. The large negative values in the Philippine Sea of (c) indicates the large "excessive" decrease of $O_3$ compared to its local consumption.

As such, we can conclude that whether the iodine-chemistry-induced $\Delta P_{OH}$ is negative or positive over the ocean is determined by the relative abundance of $O_3$ and the transport length of $O_3$-abundant air mass before it reaches the target ocean areas as the accumulation of $O_3$ decrease depends both on speed and time of the accumulation, and the latter is possibly more important (Fig. S10). In the Philippine Sea area, $O_3$ concentration is low, and $O_3$ is generally transported long away from the Philippines and the southeast boundaries that indicate the possibility to experience significant consumption by iodine when the "fresh" $O_3$ arrive there (see Fig. S10c as an example); therefore, it is possible for the dominant effect of "upwind $O_3$-depletion by iodine" along the plume paths, which results in an overall negative iodine-chemistry-induced $\Delta P_{OH}$. On the other hand, the $O_3$ in the Chinese seas and Sea of Japan have different sources and have higher concentration than that of Philippine areas, possibly resulting in a shorter transport path length before the air mass arrives ocean areas and therefore a relatively prominent effect of iodine cycling and a relatively weak effect of "upwind $O_3$-depletion by iodine" along the plume paths, which results in an overall positive iodine-chemistry-induced $\Delta P_{OH}$. In short, the negative $\Delta P_{OH}$ in part of the Philippine Sea results from the more important contribution from the effect of the $O_3$ depletion (local+upwind) than the influence of the additional local production of OH from the HOI photolysis.

### 3.4.4 Summary of the influences of the factors

In summary, we can conclude that the marine-emitted halogen species can result in a complicated change of $P_{OH}$ in East Asia (Fig. 2), with negative $\Delta P_{OH}$ in most areas of the ocean while positive in the north and south parts especially in the nearshore areas. IRR analysis results show that the changes of photolysis of $O_3$ and HOX, and the reaction $HO_2+Y$ are the main contributors to $\Delta P_{OH}$ (Fig. 3). These three pathways are influenced by different factors related to different species. For the photolysis of $O_3$, both SSA and iodine can significantly decrease it, but with different mechanisms. SSA mainly influence the photolysis through a physical factor, by extinction of solar radiation, which in turn can decrease both $J(O^1D)$ and $O_3$ concentration. Inorganic iodine can only decrease $O_3$ concentration, through the enhanced $O_3$ deposition and the atmospheric destruction of $O_3$, but to a much larger extent than that caused by SSA. For the photolysis of HOX, only the cycling of HOI has a significant contribution. For the conversion of $HO_2$ to OH, IO will compete with NO and $O_3$ to consume $HO_2$, resulting a significant decrease of the conversion; while the SSA can also lead to a decrease of the conversion, probably through more complicated feedbacks.

By influencing the strengths of the three pathways, these factors determine the pattern of the net $\Delta P_{OH}$. More specifically, the basic pattern of $\Delta P_{OH}$, with the largest relative decrease in the Philippine Sea (Fig. 2b), is controlled by the atmospheric iodine chemistry which shows negative $\Delta P_{OH}$ value only there (Fig. 6d), while the other marine areas can be roughly seen as the competition between the relatively evenly distributed negative $\Delta P_{OH}$ and the positive iodine-chemistry-induced $\Delta P_{OH}$. In the Philippine Sea, the $O_3$ concentration decrease more than that is consumed locally, resulting in an "excessive" decrease of $O_3$ and therefore negative $\Delta P_{OH}$. The excessive decrease of $O_3$ illustrates the effect of accumulated "upwind $O_3$-depletion by

iodine", which results from the high concentration of $O_3$ and possibly long transport path length of $O_3$-abundant air mass before reaching the location; whether the emission rates of HOI/$I_2$ in the area are low or high is not important. Other areas show more influence of local chemical consumption of $O_3$ (Fig. 9c), which is generally accompanied by the more efficient HOI cycling (and OH production), and therefore positive iodine-chemistry-induced $\Delta P_{OH}$.

In the Bohai Sea, the chemistry of SSA Cl also plays a role in increasing $P_{OH}$, probably due to the higher concentrations of NO$x$ (for Cl activation) and VOCs (for $RO_2$ production) there. Similarly, the NCP also shows noticeable contribution of SSA Cl chemistry on $\Delta P_{OH}$, but the impact is shaded by the negative contribution of SSA extinction effect on $\Delta P_{OH}$ (Figs. 4a, 5a,e).

## 3.5 Limitations

There are several limitations in our investigation. Our results rely much on the current halogen chemistry in CTMs, which is still under development. The uncertainty in the Cl activation and its oxidations of VOCs may have larger impacts, but the recent update of $N_2O_5$ uptake in China does not improve Cl chemistry significantly (Dai et al., 2020), and due to the complexity of VOCs reactions, there are very few studies focused on the updates of Cl-VOCs chemistry. More studies on the parameterization of Cl activation and following Cl-VOCs reactions are needed. Recent GEOS-Chem studies improved the uptake of HOBr substantially, but the two major revisions have opposite effects on BrO: increasing HOBr uptake by using more sophisticated parameterizations (Schmidt et al., 2016) and decreasing $Br_2$ yield by adding competition reactions of HOBr with S(IV) (Zhu et al., 2019;Chen et al., 2017;Wang et al., 2021). The uptake of HOBr in our study is simply parameterized with a constant reactive uptake coefficient 0.1 (Sarwar et al., 2019), which may result in a lower debromination rate. However, since Br chemistry influences OH mainly through the consumption of $O_3$ (Stone et al., 2018), the constraining on modelled BrO is sufficient for our purpose. As shown in 3.1, modelled BrO is comparable to previous studies (Zhu et al., 2019), indicating the update of HOBr chemistry may not be critical to our results, but more measurements of BrO with seasonality information are needed for further evaluation of the impacts of Br chemistry. Furthermore, the fact that exclusion of SSA debromination through HOBr+Br$^-$ reaction in several previous GEOS-Chem studies (Sherwen et al., 2016;Stone et al., 2018;Schmidt et al., 2016) does not decrease BrO burden (Zhu et al., 2019) indicates that there are more complicated interactions between different reactive bromine species ($Br_y$) and more careful check are needed. The largest limitation comes from the iodine chemistry because it is the main contributor to the change of $P_{OH}$ through different pathways. A recent observation study reported a much faster uptake of HOI and release of ICl and IBr (Tham et al., 2021), which may have large impacts on the cycling of HOI. In particular, since the photolysis of ICl and IBr is faster than that of HOI, iodine atom would be more rapidly recycled and $O_3$ would be more efficiently consumed (Tham et al., 2021) but without producing OH. At the same time, OH production from HOI photolysis would be slower since HOI is more efficiently removed from the system. As a result, the impact of iodine chemistry on OH would be more negative (Kanaya et al., 2002). Related to the iodine chemistry, the enhancement of $O_3$ deposition to the ocean is also not satisfactorily

parameterized (Loades et al., 2020;Luhar et al., 2018;Pound et al., 2020). Therefore, the incomplete halogen chemistry may limit the representativeness of our results but probably result in a larger impact of halogen chemistry on OH.

On the other hand, the uncertainties in the emission estimations cannot be fully described by using different emission rates, since some discrepancy could be driven by spatial variability of emissions. For example, a new dataset of gridded iodide concentration produced using machine learning methods based on observations has recently become available (Sherwen et al., 2019), showing different average concentration and spatial distribution from the two parameterizations used in this study (Chance et al., 2014;MacDonald et al., 2014). Future studies focusing on the impacts of iodine chemistry should better include the new dataset though the reported iodide values by Sherwen et al., (2019) lie between those calculated values used in this study. In addition, the emissions of halocarbons are less understood than SSA and inorganic iodine, and simply scaling the emission rates using global annual fluxes may not well capture the variations of emission rates in different areas. In our domain, because the observations are very sparse, the constraints on the emission estimations are very weak, and more studies are needed to better characterize the halocarbon emissions, especially in the tropical western Pacific which is potentially important for stratospheric injection.

## 4. Conclusions

To examine the impacts of gas/particle exchange between ocean and atmosphere on the regional air oxidation capacity, we explore the impact of marine-emitted halogen species on atmospheric OH in East Asia in summer. The net $\Delta P_{OH}$ caused by all marine-emitted halogen species has both positive and negative signs in the marine atmosphere and the positive $\Delta P_{OH}$ appears mainly at nearshore areas. The monthly $P_{OH}$ is generally decreased over the ocean with maxima of 10–15% in the Philippine Sea, but is increased in many nearshore areas, with maxima of 7–9% in the Bohai Sea. In the coastal areas of southern China, the monthly change of $P_{OH}$ can be comparable to or even larger than that over the ocean, though the relative values are relatively small (up to 3–5%) due to the large absolute value over land. These results indicate a notable impact of marine-emitted halogens on atmospheric oxidation capacity, which could have significant implication in the lifetime of long-lived species such as $CH_4$ (one of the major greenhouse gases) in long-term and amount of air pollutants such as $O_3$ in the episodic events.

IRR analysis show that the net effect of $\Delta P_{OH}$ is controlled by the competitions of three main pathways through different halogen species, while the contributions of other pathways are minor. In additional to the two well-known pathways involving the changes of the photolysis of $O_3$ and HOX, the competition on $HO_2$ of XO with NO and $O_3$ also significantly changes the OH production rate. These three main pathways are influenced by different factors that are related to different halogen species. SSA and inorganic iodine gases have the most significant impacts on $P_{OH}$. In this study, in additional to the chemical impacts, the physical impacts of the marine-emitted halogens on OH are also explicitly and quantitatively examined. More specifically, SSA and inorganic iodine decrease $P_{OH}$ through physical processes including the extinction effect of SSA and the enhancement of ozone deposition by oceanic iodine, while generally increasing $P_{OH}$ through chemical

processes among which the Cl (from SSA) and inorganic iodine chemistry are the most important. The physical impacts are quite comparable to the chemical impacts. In the continent, SSA is the controlling species and its extinction effect leads to the negative $\Delta P_{OH}$ in southern China. While in the ocean atmosphere, inorganic iodine gases are more important as the complicated iodine chemistry determines the basic pattern of $\Delta P_{OH}$. It is the competition between iodine's enhancing effect on the conversion of $HO_2$ to OH and iodine's decreasing effect on OH production from $O_3$ that determines the sign of iodine-chemistry-induced $\Delta P_{OH}$. The relative strengths of these two opposing effects are controlled by the $O_3$ concentration and the transport path length of $O_3$-abundant air mass over the ocean, which determine the relative importance of accumulated upwind "$O_3$-depletion by iodine" (negative effect) compared to local iodine cycle (positive effect).

Although the uncertainties in estimating the emission rates of different halogen species could influence on the magnitude and even the distribution of the halogen-induced change of $P_{OH}$, the response of the main contributors of $P_{OH}$ to the individual species and pathway and their influencing factors have been quantified, which explains the spatial variability of halogen-induced $\Delta P_{OH}$ over East Asia and also can be applied in other circumstances (e.g., different domains, regions, and emission rates).

**Data availability**

Hourly $O_3$ concentration data could be obtained from https://quotsoft.net/air/ (Ministry of Environmental Protection of China). NCEP datasets are available at https://rda.ucar.edu. The chl-*a* data can be downloaded from the merged products of the GlobColour data set (http://globcolour.info).

**Author contribution**

YL designed the study and wrote the manuscript. SF run the simulations, conducted analyses, and wrote the manuscript.

**Competing interests**

The authors declare that they have no conflict of interest.

**Acknowledgment**

We thank the editor and three anonymous reviewers for their valuable comments. We thank the principal investigators of the AERONET sites used in this study for maintaining their stations. We thank Yousuke Sawa for maintaining the Yonaguni station. This research was funded by Guangdong Basic and Applied Basic Research Fund Committee (2020B1515130003), Key Special Project for Introduced Talents Team of Southern Marine Science and Engineering Guangdong Laboratory (Guangzhou), grant number GML2019ZD0210; National Natural Science Foundation of China,

grant number 41961160728, 41575106, 42105124; Shenzhen Science and Technology Program, grant number KQTD20180411143441009, Key-Area Research and Development Program of Guangdong Province, grant number 2020B1111360001, Shenzhen Key Laboratory Foundation (ZDSYS20180208184349083), the Guangdong Province Science and Technology Planning Project of China, grant number 2017A050506003, Center for Computational Science and Engineering at Southern University of Science and Technology.

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
