# Peer review of "The impacts of marine-emitted halogens on OH radicals in East Asia during summer"

_Atmospheric Chemistry and Physics, 2021_

## Author Response (AR1)

Re: acp-2021-876 (The impacts of marine-emitted halogens on OH radical in East Asia during summer).

Dear editors,

We are grateful to the editors and the reviewers for the comments and suggestions. We have added following major modifications to address the reviewer's comments:

(1) We add more discussion about iodine chemistry to clearly revel the driving factor on the spatial impacts on  $P_{OH}$ . In this context, the interaction of anthropogenic emission and iodine chemistry is discussed.

(2) Add more discussion about the limitations in the context of guiding further development of chemical reactions in models and further observations.

Following is a point-by-point response to the reviewer's comments. Texts in *Italic Bold* are the reviewers' comments, and those in normal black are our responses. The blue texts are revised sentences in the revised manuscript. All the line numbers in blue are referred to the change tracking version. We hope that you and the referees will find the changes satisfactory and we are looking forward to hearing from you soon.

\_\_\_\_\_

**Responds to the reviewers' comments**

**Reviewer #1:**

**Major:**

1. How would anthropogenic pollutants interact with marine halogens? The authors have cited a wide range of previous studies (e.g., Sherwen et al., Wang et al.,) but most of these studies focus on the global scale where halogen chemistry is largely driven by natural processes. In this work, the studied area is subject to major anthropogenic influence. This is a question of great interest that many global models are difficult to address because of the coarser resolution. I would suggest that the authors expand on this, which places this work in the context of previous studies and greatly advances our current understanding of reactive halogen chemistry.

**Response:**

Thank you very much for your great comment. In previous study, the dependence of the activation of SSA Cl on anthropogenic pollutants NOx have been widely discussed, so we do not discuss it again in this work. What we focus on is the anthropogenic NOx and VOC pollutant influence on the reactive halogen chemistry. We add new simulations increasing ozone level by enlarge of NOx and VOCs emissions in the Philippines, where show very special negative change of  $\Delta P_{OH}$ , to illustrate the impact of anthropogenic emission influence on the iodine-chemistry-induced  $\Delta P_{OH}$ . The added content can be referred to the line 514-558 in the revised manuscript.

2. Figure 2 shows some striking discrepancies, which are, again, very intriguing. Section 3.3 and 3.4 are able to identify several processes that are probably less important. However, these sections fail to offer a clear explanation on what exactly drives the spatial variability of iodine-induced OH production. I have a few detailed suggestions in the next section to improve the clarity. In addition, I would also suggest that the authors provide a few more maps showing the surface seawater iodide field, the modeled marine emissions of I2 and HOI, the modeled ozone dry deposition velocities, as well as the modeled HOI and sea salt aerosols in the surface air. Perhaps these can shed insights into the spatial variability of iodine-induced OH production in this region.

**Response:**

Thank you very much for your great comment. We agree that driving factor for the contrast pattern of  $P_{OH}$  change, especially the negative change of  $P_{OH}$  in the Philippines Sea, is worthy to be explained further. We add spatial maps as suggested and further discussions to explain the discrepancies in the revised manuscript (Line 481-514).

**Specific comments:**

**1. Line 15: Please clarify how monthly OH production (P\_OH) is calculated. Does this involve nighttime signals?**

**Response:**

Yes, it is the average of  $24 \times 31$  hourly POH. This is in line with previous global studies where only global mean OH is discussed (e.g., Sherwen et al. 2016a; Stone et al. 2018). Daytime average will be explicitly mentioned if it is the case.

2. Line 33: This is a vague description. HO2 -> OH is not always a net source of OH, since a major fraction of HO2 is actually produced from OH. Please rewrite this sentence: either tease out the primary fraction of HO2 (e.g., from formaldehyde) or discuss the sum of OH and HO2. After all the interconversion between OH and HO2 is fast.

**Response:**

Thanks for the suggestion. We modify the manuscript as follows:

When there is abundant NO, as typically in the polluted continental atmosphere, peroxy radicals (RO2, and HO2) will be formed by the oxidation of hydrocarbons by OH, and will form OH again in the reaction with NO. This HOx (=OH+HO2) cycling maintain a high OH concentration that cannot be achieved by primary sources alone. (Line 48-51)

3. Line 37: ... under low-NOx condition. Also please clarify what exactly is "low-NOx condition". Is this referring to the condition when RO2 fate is dictated by non-NOx pathways? If this is the case, often the NOx threshold can be very low.

**Response:**

Yes, it is. Low-NOx condition is actually the very low NOx threshold. To avoid misleading, we modify the manuscript as follows:

Due to the complexity of the HOx chemistry, the sources and sinks of OH are not fully understood. For example, recent studies showed that when NOx concentration is very low there may be missing sources of OH. (Fittschen et al. 2019; Fuchs et al. 2013; Hofzumahaus et al. 2009; Lelieveld et al. 2008; Lu et al. 2019; Rohrer et al. 2014; Stone et al. 2012; Tan et al. 2019; Whalley et al. 2021). Besides, HOx chemistry can interact with other oxidizers in the atmosphere in specific circumstances. (Line 51-59)

**4. Line 40-43: The end of this paragraph is confusing and unclear how this is relevant for this manuscript.**

**Response:**

Thanks. It has been removed from the revised manuscript. Please see the response to comment #3 above.

**5. Line 46: ... under high NOx condition.**

**Response:**

Thanks. It has been removed from the revised MS. Please see the response to comment #3 above.

6. Line 50: What is "long-term species"? I assume what the authors meant to say is "long-lived" since ozone lifetime is longer than, say, HOx radicals? If this is the case, this is clearly not true since ozone is not always constrained in box models. For instance, there are numerous studies focusing on halogen-induced ozone destruction in the Arctic and the combined effects on HOx and they use box model with ozone unconstrained.

**Response:**

Thank you. Yes, we mean "long-lived". We agree that ozone can be unconstrained

in box models. In the Arctic, the focus is usually the ozone depletion events and ozone is indeed unconstrained. But many related studies do constrain O3.

We modify the sentence from "In box-model studies, long-term species such as O3 are usually observation-constrained and cannot reflect the influence of halogens, probably resulting the difference between box models and CTMs (Stone et al. 2018)." to "In a box model, when the long-lived species such as O3 are observation-constrained, it cannot reflect the complete influence of halogens, which probably result in the different results between box models and CTMs (Stone et al. 2018)." (Line 88-91)

7. Line 45: "One relevant reaction is that XO (X=Cl, Br, and I) transform HO2 to OH..." This is very confusing as written since this is not a one-step process: HOX is produced first, which may undergo photolysis and produce OH but HOX can also undergo heterogeneous uptake on aerosols (a major driver of halogen cycling).

**Response:**

We agree that HOX may undergo heterogeneous uptake and activate particulate halogens but the rate of HOX uptake is much slower than the photolysis of HOX and almost all HOX would be photolyzed to form OH; therefore, the main result of XO+HO2 is to transform HO2 to OH, which has been adopted by many authors to explain the HOx cycle in the presence of XO. For example, (Saiz-Lopez and von Glasow 2012) stated that "In the presence of reactive halogens, the HOx balance is shifted towards OH." To make it clearer, we rephrase the sentence from "XO (X=Cl, Br, and I) transform HO2 to OH, as NO does in the high-NOx condition." to "XO (X=Cl, Br, and I) shifts the HOx balance towards OH (Saiz-Lopez and von Glasow 2012)." (Line 65-66)

8. Line 54: "... but it is not very clear which process will dominate" Respectfully, I disagree. Perhaps this is not explicitly spelled out in some studies, but the final model outcome speaks for itself. For instance, Wang et al., (2021) showed that the net effect of halogen chemistry on global tropospheric HOx is that both OH and HO2 are reduced by 3-4%. This is buffered by many other processes but the primary driver is a global ~10% decrease in HOx production from ozone. This is qualitatively consistent with previous studies.

**Response:**

We agree that the sentence is somewhat misleading, and we delete the sentence "but it is not very clear which process will dominate" and modify it as follows:

Even though we know all the important pathways, due to their opposite impacts on OH, we need further to understand the controlling processes of these pathways in order to better explain the trend of halogen-induced  $\Delta$ OH in a specific circumstance. (Line 99-101)

9. Line 56-57: "For example, the conversion of HO2 to OH enhanced by XO would consume HO2, which in turn should decrease the conversion through HO2+NO.

Previous CTM studies generally did not consider such an impact" I am confused. These two are competing processes with the same end goal, which is to convert HO2 to OH. As long as these relevant mechanisms are included in the model, the impact will be considered.

**Response:**

Sorry for the confusing. The case is that all the relevant reactions are included in the model in previous study. They could be considered, but not yet been considered in the result explanation. We have deleted this sentence in the revised MS.

These two (HO2+XO and HO2 +NO) are processes with the same end. Our point is that HO2+NO and HO2+XO have different efficacy in converting HO2 and are differently influenced by the environment. With the competition of XO with NO, the conversion of HO2 to OH may be changed. That is why HO2 may not be converted more to OH by Cl and Br (and only I enhance the conversion of HO2 to OH significantly) as shown by the negative change of  $P_{OH_HO2} + P_{OH_HOX}$  in Fig. R1.

**Fig. R1** (Fig. S4b). Change of  $P_{OH_HO2} + P_{OH_HOX}$  caused by SSA Br chemistry (=case with SSA Br emission minus case without).

**10. Line 58-59: I failed to follow how exactly halogen chemistry can affect the photolysis of HONO and H2O2, ozonolysis of some alkene, ... Please clarify.**

**Response:**

Halogen chemistry has very complicated influence on HOx chemistry because there are lots of chemical feedbacks. HONO, for example, is primarily a product of heterogeneous reactions in CMAQ, which will consume NO2. As NOx is complicated influenced by halogen chemistry, the potential impacts on HONO cannot be excluded when we want to thoroughly quantify the impacts of halogen chemistry on HOx chemistry. We tracked all OH sources change under the impact of halogens. These three are chosen for illustrations. Of course, Fig. 2 just prove the unimportance of the halogens' impacts on these OH sources, but this is a result, not a premise.

11. Line 63: "However, previous studies did not analyze the pathways..." Again I disagree with this statement. Globally, iodine chemistry alone may have largely compensating effect on OH (e.g., Sherwen et al., 2016) but the relative abundance of reactive iodine species and the impacts of iodine chemistry on the global tropospheric OH levels in the context of chlorine and bromine chemistry is shown in Wang et al. (2021): globally, the effect of halogen chemistry (including iodine) on OH, is a net reduction. The relative importance of iodine chemistry on the global scale thus can be inferred.

**Response:**

We revise the sentence. Please see comment #8 above.

12. Line 65-66: This is very vague as written. Do the authors refer to the reactive halogens (e.g., 12/HOI), debromination from sea salt, or the very short-lived substances (VSLS)? Please clarify. But generally this is a valid point, and more recent estimates certainly benefit from more comprehensive observations, thus yielding narrower ranges compared to earlier studies. Either way, a few representative citations are warranted. I'll list a few more recent studies for each broad topic for the author's consideration: Iodide-driven 12/HOI emission and ozone deposition (Carpenter et al., 2021; Chance et al., 2019; Inamdar et al., 2020; Karagodin-Doyennel et al., 2021; Pound et al., 2020; Sherwen et al., 2019; Wang et al., 2021; sea salt debromination (Zhu et al., 2019); VSLS (Lennartz et al., 2015; Ordóñez et al., 2012; Wang et al., 2019; Ziska et al., 2013).

**Response:**

Thanks for your comment. Yes, it is. We revise the sentence as follows:

Moreover, since current estimations of marine halogen emissions, including SSA Cl and Br ions (and their activations), inorganic iodine (I2 and HOI), and very short-lived halocarbons), have large uncertainties (Carpenter et al., 2021;Ordóñez et al., 2012;Ziska et al., 2013;Inamdar et al., 2020;Lennartz et al., 2015;Zhu et al., 2019;Sekiya et al., 2020;Wang et al., 2021;Grythe et al., 2014). (Line 101-104)

Thanks for the useful reference information. We described our calculations of halogen emissions in section 2.2 of the manuscript.

13. Line 135-136: please show the original and scaled halocarbon emissions. Please note that although this is a wildly use approach (i.e., scale to chlorophyll-a), there is no robust relationship between many VSLS and chlorophyll-a (Carpenter et al., 2009; Chance et al., 2014; Liu et al., 2013)

**Response:**

The original halocarbon emissions are based on that of Ordóñez et al. (2012). Please see table R1 for the scale factors.

**Table R1(S2)**. Global annual fluxes of very short-lived halocarbons reported in previous studies and scale factors of halocarbon emissions used in this study.

| Species                         | Annual flux (Gg/yr)   |         |          |      | Scale factor |  |
|---------------------------------|-----------------------|---------|----------|------|--------------|--|
|                                 | Ordóñez et al. (2012) | WMO low | WMO high | low  | high         |  |
| CHBr 3               | 533                   | 126     | 865      | 0.24 | 1.62         |  |
| CH 2 Br 2 | 67.3                  | 62      | 109      | 0.92 | 1.62         |  |
| CH₃I                            | 303                   | 176     | 615      | 0.58 | 2.03         |  |
| CH 2 BrCl            | 10.0                  | 6.48    | 9.72     | 0.65 | 1            |  |
| CHBr 2 Cl            | 19.7                  | 19.6    | 56.1     | 1    | 2.85         |  |
| CHBrCl 2             | 22.6                  | 16.4    | 22.6     | 0.73 | 1            |  |
| CH 2 ICI      | 234                   |         |          | 0.58 | 2.03         |  |
| CH 2 IBr             | 87.3                  |         |          | 0.58 | 2.03         |  |
| CH 2 I 2  | 116                   |         |          | 0.58 | 2.03         |  |

We agree that the estimation of VSLS emission is very rough.

**14. Line 139: the iodide-driven ozone deposition is coupled with the reactive iodine emission in several recent studies (Karagodin-Doyennel et al., 2021; Pound et al., 2020).**

**Response:**

We thank the reviewer for the references. We are aware of such kind of "couple", which may be called apparent couple where the two processes are related only through  $O_3$  concentration.

By "couple" we mean the actual couple of the two processes through the aqueous reactions in the sea surface. It is the aqueous reactions that consume O3 and produce inorganic iodine at the same time. This approach is conducted to develop the parameterization of I2/HOI emission by Carpenter et al. (2013) (see their Table

S1).

To avoid confusion, we delete "but current CTMs do not couple these two processes" in the revised manuscript (Line 187-188).

15. Line 142: Sherwen et al., (2019) showed that the McDonald et al. iodide parameterization underestimates the surface seawater iodide by roughly a factor of 2 on the global scale. Chance et al. is improved but shows wider variability compared to observations.

**Response:**

Thanks for the comment. Since the reported iodide values by Sherwen et al., (2019) lie between those calculated values according to parameterization of McDonald et al. (2014) and Chance et al. (2014), we chose the latter two to conduct sensitivity simulations.

16. Line 150-: The amount of ozone measurements used to demonstrate the performance of this model (Figure S1) is remarkable. But this is less relevant for this study since the majority of the stations are located inland and hence are probably not heavily impacted by the marine halogens. What is directly relevant for this study is, the modeled ozone levels in those sensitivity studies, especially in the All\_High case, and how would these compare to the coastal ozone measurements when the air masses are primarily originated from the ocean.

**Response:**

Thank you very much for you great comment. We realize that the modeled ozone level is important and add further discussion about it. Please see the response to major comment #2 above. Besides, we add a comparison of simulated and observed O3 in an island in the Philippine Sea. Please see Fig. S1b in the revised manuscript (Fig. R2 below).

---

## Author Response (AR2)

Re: acp-2021-876 (The impacts of marine-emitted halogens on OH radical in East Asia during summer).

Dear editors,

We are grateful to the editors and the reviewers for the comments and suggestions. Following is a point-by-point response to the reviewer's comments. Texts in ***Italic Bold*** are the reviewers' comments, and those in normal black are our responses. The blue texts are revised sentences in the revised manuscript. All the line numbers in blue are referred to the change tracking version. We hope that you and the referees will find the changes satisfactory and we are looking forward to hearing from you soon.
* * *
**Responds to the reviewers' comments**

**Reviewer #1:**

***This is the second round of review of the manuscript entitled "The impacts of marine-emitted halogens on OH radicals in East Asia during summer" by Shidong Fan and Ying Li. It is my opinion that the revised manuscript has addressed most of the major issues identified in the original manuscript, and the quality, especially the clarity, of the manuscript is greatly improved. Especially, the major drivers of iodine-induced OH production are clearly identified via Figures 3, 4, and 7, and Figure 8 made some of the key inputs transparent. The authors also carried out additional sensitivity tests (in addition to the nicely designed suite of experiments) to evaluate the impact of anthropogenic emissions on halogen chemistry. I appreciate the efforts the authors have invested in addressing my previous comments. I would recommend this manuscript for publication after very minor (mostly technical) suggestions and clarifications.***

***1. Line 47 (revised manuscript): "… that when NOx concentration is very low…" Please specify how low is considered as low.***

**Response:**

"low" generally means that NO concentration below 1 ppbv, but different studies may use different thresholds (e.g., 0.3 ppbv in Tan et al. (2017) and 0.4 ppbv in Rohrer et al. (2014)). We add "(e.g., NO concentration less than several hundred pptv)" in the revised manuscript.

*2. Line 72-74: "… the stability of the interaction result of different pathways…" I am not sure what this means. Please clarify or rephrase.*

**Response:**

It means that different emission rates of different halogen species may change the net effect because different pathways are differently influenced by emissions. We change "the stability of the interaction result" to "the net effect".

*3. Line 92: Please indicate the pressure altitude of the model top.*

**Response:**

The pressure of the top layer is 50 hPa, which is added in Line 92.

*4. Line 116: I could be wrong but I don't think SSA is defined.*

**Response:**

SSA is defined in the Abstract (line 23-24). It is short for sea spray aerosols (just sea salt aerosols in our model). We define it again at the first mention in the main text now.

*5. Line 146-148: The authors tested two outdated surface seawater iodide parameterizations (Chance et al. 2014 and McDonald et al. 2014) even though a more advanced data product is available (Sherwen et al. 2019). The authors argued that "Since the reported iodide values by Sherwen et al., (2019) lie between those calculated values according to parameterization of McDonald et al. (2014) and Chance et al. (2014), we chose the latter two to conduct sensitivity simulations." Do keep in mind that the spatial distribution and dynamic range are also very different in Sherwen et al. (2019), especially in the tropical/subtropical western Pacific. I will not ask the authors to perform additional simulations at this stage, but this remains another apparent limitation of this study (both surface seawater parameterizations tested in this work are outdated) that should be noted in Section 3.5 (Limitations of this work).*

**Response:**

Thanks for the comment. We add a paragraph in the revised manuscript to mention this limitation (Line 544-554).

*6. Line 149-151: I think Ordóñez et al. (2012) parametrization essentially has constant values in the subtropics and high latitude regions (e.g., north of 20N). I am not sure if simply scaling up based on global emission makes much sense, since much of the discrepancy is driven by spatial variabilities. It would be great to compare the scaled surface seawater halocarbons to the surface seawater observations in this region (e.g., Fuhlbrügge et al., 2016, Fiehn et al., 2017). But I do realize that these halocarbons play a relatively minor role as revealed in this work. Nevertheless, it is worth mentioning this in Section 3.5 (Limitations of this work) that the halocarbons in this region (tropical western Pacific) remains poorly understood which is*

*potentially important for stratospheric injection.*

**Response:**

Thank you for the great comment. We add a paragraph to discuss the limitations in emissions (Line 544-554).

*7. Line 324: "Since these factors are generally species-related…" What factors? Please clarify.*

**Response:**

The factors have been specified in line 310-313 and presented in R1-R4. To make it clearer, we rephrased the sentence from "Since these factors are generally species-related" to "Since these factors just mentioned above are generally species-related".

*8. Line 355: Please remind the readers where the Greater Bay Area is. Consider labeling that in the map.*

**Response:**

Thank you for your suggestion. We realize that in a relatively large domain, denoting a small area may distract the attention of readers. Since this information of location is not very important, we delete "the Greater Bay Area" in the revised manuscript and only keep "southern China". Since the Greater Bay Area is in southern China, the statements about maxima or minima still hold.

Rohrer, F., K. D. Lu, A. Hofzumahaus, B. Bohn, T. Brauers, C. C. Chang, H. Fuchs, R. Haseler, F. Holland, M. Hu, K. Kita, Y. Kondo, X. Li, S. R. Lou, A. Oebel, M. Shao, L. M. Zeng, T. Zhu, Y. H. Zhang, and A. Wahner, 2014: Maximum efficiency in the hydroxyl-radical-based self-cleansing of the troposphere. *Nature Geoscience*, **7,** 559-563.

Tan, Z. F., H. Fuchs, K. D. Lu, A. Hofzumahaus, B. Bohn, S. Broch, H. B. Dong, S. Gomm, R. Haseler, L. Y. He, F. Holland, X. Li, Y. Liu, S. H. Lu, F. Rohrer, M. Shao, B. L. Wang, M. Wang, Y. S. Wu, L. M. Zeng, Y. S. Zhang, A. Wahner, and Y. H. Zhang, 2017: Radical chemistry at a rural site (Wangdu) in the North China Plain: observation and model calculations of OH, HO2 and RO2 radicals. *Atmospheric Chemistry and Physics*, **17,** 663-690.

**Reviewer #3:**

*My previous points were mostly adequately addressed and the manuscript was improved. I have the following minor points for further consideration.*

*1. I previously asked clarification of assumed single scattering albedo used to calculate J(O1D). It seems it was confused with sea spray aerosols (both with acronym of SSA) and thus excluded from clarification. I have tested quick TUV calculations and found that about 5% J(O1D) decrease could occur with Single scattering albedo = 0.95 with AOD = 0.2. I thought the marine aerosols could be brighter (single scattering albedo >0.95). But was that the range that the authors assumed?*

**Response:**

We are sorry for not clarifying single scattering albedo used in CMAQ in our last response. We used CMAQv5.3 to conduct our simulations, which does not need to pre-define single scattering albedo values. Instead, photolysis rates of $O_3$ and $NO_2$ are calculated each numerical step. Aerosol concentrations and complex refractive indices of five types of particles (WATER, SOLUTE, DUST, SEASALT, SOOT, see https://www.airqualitymodeling.org/index.php/CMAQv5.1_In-line_Calculation_of_Photolysis_Rates) are used to calculate the optical properties of aerosols. The details can be found in Binkowski et al. (2007) (and references therein).

In old versions of CMAQ (before v5.1), photolysis rates are calculated off-line using a lookup table. Single scattering albedo needs to be provided and it is 0.99 (in the version 5.0.1). Considering the continuity of the development of the model (CMAQ), we therefore infer that the single scattering albedo for sea salt would be larger than 0.95.

*2. The explanation about why the InorgI_chem can result in "negative" values over the Philippine Sea was inserted in Lines 481-558 but was a bit too lengthy. Overall, in short, the effect of the O3 level decrease (including in the upstream region) was more important than the influence of additional production of OH from the HOI photolysis?*

**Response:**

In short, yes. In the Philippine Sea, the effect of $O_3$ decrease is more important than the OH production from HOI cycling and emission. This is also the case in many other oceans because $O_3$ is generally more "aged" (see e.g., Stone et al. 2018). But in the China seas and Sea of Japan, due to the larger influence of the nearby lands, the effect of $O_3$ decrease is not fully developed and is therefore less important than the OH production from HOI.

**References**

Binkowski, F. S., S. Arunachalam, Z. Adelman, and J. P. Pinto, 2007: Examining photolysis rates with a prototype Online photolysis module in CMAQ. *Journal of Applied Meteorology and Climatology*, **46,** 1252-1256.

Stone, D., T. Sherwen, M. J. Evans, S. Vaughan, T. Ingham, L. K. Whalley, P. M. Edwards, K. A. Read, J. D. Lee, S. J. Moller, L. J. Carpenter, A. C. Lewis, and D. E. Heard, 2018: Impacts of bromine and iodine chemistry on tropospheric OH and $HO_2$: comparing observations with box and global model perspectives. *Atmospheric Chemistry and Physics*, **18,** 3541-3561.